theoretical biology, evolution, developmental biology

phenotypic plasticity, evolution, development, sensitive periods, environmental change, mathematical modelling

**Author for correspondence:**
Nicole Walasek
e-mail: walasek.nicole@gmail.com

# Sensitive periods, but not critical periods, evolve in a fluctuating environment: a model of incremental development

Nicole Walasek[1], Willem E. Frankenhuis[1,2,3] and Karthik Panchanathan[4]

[1]Behavioral Science Institute, Radboud University, 6525 GD Nijmegen, The Netherlands
[2]Department of Psychology, Utrecht University, 3584 CS Utrecht, The Netherlands
[3]Max Planck Institute for the Study of Crime, Security and Law, 79100 Freiburg, Germany
[4]Department of Anthropology, University of Missouri, USA

 NW, 0000-0003-4411-319X; WEF, 0000-0002-4628-1712

Sensitive periods, during which the impact of experience on phenotype is larger than in other periods, exist in all classes of organisms, yet little is known about their evolution. Recent mathematical modelling has explored the conditions in which natural selection favours sensitive periods. These models have assumed that the environment is stable across ontogeny or that organisms can develop phenotypes instantaneously at any age. Neither assumption generally holds. Here, we present a model in which organisms gradually tailor their phenotypes to an environment that fluctuates across ontogeny, while receiving cost-free, imperfect cues to the current environmental state. We vary the rate of environmental change, the reliability of cues and the duration of adulthood relative to ontogeny. We use stochastic dynamic programming to compute optimal policies. From these policies, we simulate levels of plasticity across ontogeny and obtain mature phenotypes. Our results show that sensitive periods can occur at the onset, midway through and even towards the end of ontogeny. In contrast with models assuming stable environments, organisms always retain residual plasticity late in ontogeny. We conclude that critical periods, after which plasticity is zero, are unlikely to be favoured in environments that fluctuate across ontogeny.

## 1. Introduction

Phenotypic plasticity—the capacity of a single genotype to produce multiple phenotypes depending on environmental and somatic conditions—is widespread in nature [1–3]. There is well-established theory exploring the conditions in which phenotypic plasticity is favoured by natural selection over non-plastic development. This work has provided valuable insights (for review, see [4]). For instance, plasticity is likely to be favoured when the environment changes between generations at a rate too fast for genetic evolution to track, but slowly enough within generations for organisms to benefit from using early experience to guide phenotypic development [4–6]. However, this work has not focused on the question of how natural selection shapes *changes in plasticity* across ontogeny for different species, individuals and traits.

### (a) Modelling sensitive periods

Recently, mathematical models have been used to explore how natural selection shapes sensitive periods, i.e. time periods or life stages during which the impact of experience on phenotypic development is greater than at other times or stages (reviewed in [7–9]). In these models, organisms typically begin ontogeny uncertain about the state of their environment and gradually reduce uncertainty by sampling environmental cues. As a result, plasticity is typically highest at

the onset of ontogeny and gradually declines. These models have mainly focused on stable environments across ontogeny [10–15] (for an exception, see [16]). However, many species and populations experience environmental fluctuations within generations as well [6]. Little is known about optimal levels of plasticity across ontogeny in such conditions. For instance, when conditions fluctuate across ontogeny, plasticity may be prolonged to enable phenotypic adjustments across all of ontogeny [10,12,17]. Such a pattern would differ from that observed in models of stable environments, which often favour critical periods, where plasticity drops to zero [17,18].

## (b) Sensitive periods in fluctuating conditions

We know of only one model that has explored the evolution of sensitive periods in an ontogenetically fluctuating environment. Fischer *et al.* [16] modelled an environment that fluctuates stochastically between two discrete states. Organisms develop initial phenotypes at the onset of ontogeny based on the long-term distribution of environments. In subsequent time periods, organisms sample imperfect cues to the current environmental state and adjust their phenotypes to maximize survival and reproduction across ontogeny. As with models of stable conditions, in this model, plasticity declines with age. However, in contrast with those models, the highest level of plasticity ('peak-plasticity') does not always occur at the onset of ontogeny. In some conditions, organisms delay phenotypic adjustment until uncertainty has been sufficiently reduced, resulting in peak-plasticity shortly after the onset of ontogeny.

The model by Fischer *et al.* [16] offered a crucial step forward but also has two limitations. First, it assumes that any phenotype can develop at any age within a single time period. This assumption does not apply when phenotypes are gradually constructed or cannot be reversed. In such cases, the current phenotypic state constrains the range of phenotypes available in the future. Second, the Fischer *et al.* model measures plasticity as phenotypic change directly following a cue. However, there are other possibilities that afford different insights [9]. For example, we can explore the effects of cues on developmental trajectories and mature phenotypes [19], matching commonly used empirical designs [20,21].

## (c) Our contribution

Here, we present a model in which traits develop incrementally in an environment that fluctuates across ontogeny, exploring how cues shape plasticity across ontogeny and adult phenotypes. Organisms that gradually tailor phenotypes cannot instantaneously develop any phenotype at any time. Such incremental development is widespread in nature. For instance, plants gradually develop leaf morphology, such as area, thickness and dissection, in response to light intensity, humidity and temperature [22–24]. Animals develop morphological defenses, such as protective armour, increased body size or longer tails, as well as changes in coloration, in response to predator cues [25]. In humans, the development of motor skills appears stepwise if measures are taken across weeks or months. However, this pattern reflects smaller incremental changes, which are visible once measures are taken frequently on shorter time scales [26].

In our model, the environment varies stochastically between two discrete states. Organisms incrementally construct phenotypes while sampling cost-free, imperfect cues to the current conditions. Once phenotypic increments have developed, they cannot be undone. We use stochastic dynamic programming to compute optimal policies for a range of environments, varying the rate of environmental fluctuations, the reliability of cues, and how long adulthood lasts relative to ontogeny. These policies specify the optimal decision for each possible state of an organism, depending on its current phenotype and cues sampled. We then simulate populations of organisms following the optimal policy. Finally, we use experimental designs, matching those used in empirical studies, to quantify plasticity across ontogeny and distributions of mature phenotypes.

## 2. Model

### (a) Evolutionary ecology

The environment consists of an infinite number of discrete and non-overlapping patches. Each patch can be in one of two states, $E_0$ or $E_1$. From one time period to the next, the state of each patch switches stochastically between $E_0$ and $E_1$ with transition probabilities, $P(E_{0,t} | E_{1,t-1})$ and $P(E_{1,t} | E_{0,t-1})$, where $t$ denotes the current time period. For example, a patch might start out rich in one food type and switch to a different food type (e.g. seeds or fruits). We use a Markov process to fully describe the transitions between states. As per time period transition probabilities are fixed, we abbreviate $P(E_{0,t} | E_{1,t-1})$ and $P(E_{1,t} | E_{0,t-1})$ with $P(E_0 | E_1)$ and $P(E_1 | E_0)$.

We explore symmetric, $P(E_0 | E_1) = P(E_1 | E_0)$ and asymmetric transition probabilities, $P(E_0 | E_1) \neq P(E_1 | E_0)$. We also vary how likely transitions occur, ranging from 0.1 to 0.45 (positive autocorrelation). At the low end, the environment is relatively 'stable': an environmental switch is unlikely to occur. At the high end, the environment is 'unstable': a switch is almost as likely as no switch. We do not explore transition probabilities of 0.5 or larger (negative autocorrelation).

### (b) Phenotypic development

Organisms are born, randomly disperse into a new patch, develop to maturity in the new patch, reproduce and die. Ontogeny, $T_{ont}$, is fixed at 10 discrete and non-overlapping time periods. We obtain similar qualitative results for a larger number of time periods (electronic supplementary material, S1, figures E1.1–E1.4). We vary the length of adulthood ($T_{adult} = 1$, 5, and 20 time periods) to explore different ratios of adulthood to ontogeny (see electronic supplementary material, S2, figures E2.1–E2.4 for an adult lifespan of 10 time periods). Thus, time runs from $t = 0$ (birth) until the end of the reproductive phase $T_{end}$, such that $T_{end} = T_{ont} + T_{adult}$.

For each environmental state, there is a corresponding optimal phenotype: $P_0$ for $E_0$ (e.g. specialized for foraging seeds) and $P_1$ for $E_1$ (e.g. specialized for foraging fruits). These phenotypes represent two different traits, rather than a single trait that increases or decreases. In other words, phenotypes are not arrayed along a single dimension, but along two independent dimensions. Changes in one trait are independent of changes in the other trait. Organisms learn about their environment by receiving cost-free,

imperfect cues. After each cue, organisms have three options: specialize one increment towards $P_0$, specialize one increment towards $P_1$ or wait and forgo specialization. Once an increment has developed, it cannot be undone, yet organisms may always switch developmental trajectories.

In adulthood, organisms experience the same transition probabilities as during ontogeny, but cannot adjust phenotypes. Instead, they accrue fitness depending on the phenotype–environment fit and reproduce proportional to fitness. In this model, fitness is only a function of fertility. We consider the effects of viability selection in §5.

## (c) Learning about the environment

The organism is adapted to the transition probabilities between states, as well as the long-term probability distribution over states (i.e. the stationary distribution of the Markov process), denoted by $\pi(E_0) = P(E_0 \mid E_1)/(P(E_0 \mid E_1) + P(E_1 \mid E_0))$ and $\pi(E_1) = 1 - \pi(E_0)$. This distribution serves as an organism's evolutionary prior of being in each of the two environmental states at the onset of ontogeny [27]. If transitions towards the seed-rich state are more likely than transitions towards the fruit-rich state, i.e. $P(E_1 \mid E_0) > P(E_0 \mid E_1)$, the long-term probability of being in the seed-rich state is higher than that of being in the fruit-rich state, i.e. $\pi(E_1) > \pi(E_0)$. Symmetric transition probabilities produce a uniform stationary distribution, i.e. $\pi(E_0) = \pi(E_1) = 0.5$, while asymmetric transition probabilities produce a non-uniform stationary distribution, where $\pi(E_0) > \pi(E_1)$ if $P(E_0 \mid E_1) > P(E_1 \mid E_0)$.

In each time period, organisms sample a cost-free, imperfect cue to the current state of the environment and update their estimates according to Bayes' theorem [14,28–30]. The cue reliability is defined by the conditional probability of sampling the correct cue in the corresponding state, $P(C_0 \mid E_0) = P(C_1 \mid E_1)$. The probability of sampling an incorrect cue corresponds to $P(C_1 \mid E_0) = 1 - P(C_0 \mid E_0)$ and $P(C_0 \mid E_1) = 1 - P(C_1 \mid E_1)$, respectively. We vary the cue reliability from low (0.55) to high (0.95). The higher the cue reliability, the better organisms can adjust to the current state of the environment and exploit positive autocorrelation to adjust to future states of the environment.

## (d) Fitness during adulthood

Organisms that wait and forgo specialization across all of ontogeny attain a baseline fitness. Any developed specializations lead to increases or decreases from this baseline. During each time period in adulthood, fitness depends on the current phenotype–environment match. Total fitness corresponds to the sum of the fitness scores across adulthood.

We consider phenotypic specializations matching the environmental state as 'correct' and specializations towards the other state as 'incorrect'. We assume that correct phenotypic specializations increase fitness and incorrect ones decrease it relative to baseline fitness [31]. The fitness in each period of adulthood is calculated by summing the marginal rewards for correct specializations, marginal penalties for incorrect ones and baseline fitness. We explore three mappings between phenotypes and marginal fitness rewards and penalties (linear, increasing and diminishing) and three penalty weights (0.5, 1 and 2) [11,12,32]. The specific combination of mappings and penalty weight determines how organisms accrue fitness. Returning to our example of seed

and fruit specialization, imagine the following two organisms: organism A has developed equal numbers of specializations for both states, while organism B has waited throughout ontogeny, developing zero specializations for either state. If we assume linear reward and penalty functions and a penalty weight of 1, then both organisms accrue zero fitness. If, instead, we assume a higher penalty weight or a diminishing penalty function, then, all else equal, A would attain lower fitness than B. With a lower penalty weight or an increasing penalty function, B does better than A. In this paper, we set the penalty weight to 1 and the reward and penalty mappings to linear. We present the other combinations in the electronic supplementary material, (electronic supplementary material, S7 and S8) and address them in §5.

We describe fitness functions and formulae of all mappings in electronic supplementary material, S3.

## (e) Optimal developmental policies

To obtain optimal policies, we use the posterior estimates across ontogeny to compute expected fitness across adulthood. We treat the states of the environment during ontogeny as 'hidden', unobserved states and sampled cues as 'observed' states of a Hidden Markov Model. We then apply the forward algorithm to compute the posterior probabilities, $P(E_0 \mid D_t)$ and $P(E_1 \mid D_t)$ for all possible orderings of sampled cues $D_t$ [33]. $D_t = \{x_1, x_2, \ldots x_t\}$ denotes the sequence of cues until time period $t$, where $x_1$, $x_2$, and so forth until $x_t$ denote the cue ($C_0$ or $C_1$) sampled in each time period. We provide the formula of the forward algorithm in electronic supplementary material, S3.

In contrast with ontogeny, we model adulthood as a Markov Model with environmental states as observed states and no hidden states. The probabilities of starting adulthood in $E_0$ or $E_1$ equal the posteriors in the final time period of ontogeny. We compute the probabilities of being in each of the two states across adulthood, $P(E_0)$ and $P(E_1)$, based on these posteriors and the transition probabilities. Then, we use $P(E_0)$ and $P(E_1)$ to compute expected fitness across adulthood (electronic supplementary material, S3).

Finally, we compute optimal developmental policies using stochastic dynamic programming via backwards induction (electronic supplementary material, S3). The algorithm uses the posterior probabilities at the end of ontogeny and the expected fitness across adulthood as a starting point to determine the optimal decision in the final time period and then works its way backwards in time. All code is written in Python 2.7. and available on GitHub (https://github.com/Nicole-Walasek/SensitivePeriodsInFluctuatingEnvironments).

# 3. Analyses

## (a) From transition probabilities to autocorrelation

Empirical studies often use temporal autocorrelation to measure environmental change. To facilitate comparisons between our model and such studies, we compute autocorrelation values from transition probabilities (electronic supplementary material, S4). Higher transition probabilities produce lower autocorrelations.

The magnitude of the difference between $P(E_0 \mid E_1)$ and $P(E_1 \mid E_0)$, the 'asymmetry', also affects the autocorrelation. Suppose transitions to one state are more likely than to the

other, for example, $P(E_0 | E_1) = 0.1$ and $P(E_1 | E_0) = 0.2$. In this case, the asymmetry is 0.1. If the patch starts in $E_1$, transitions are initially quite unlikely. However, once the state switches, the probability of another switch is higher. Overall, there would be more switches and lower autocorrelation compared to a scenario in which $P(E_0 | E_1) = P(E_1 | E_0) = 0.1$. Higher asymmetries thus imply a smaller range of autocorrelations. Different sets of transition probabilities and asymmetries can approximate the same autocorrelation (see electronic supplementary material, figure E4.1).

We have explored different asymmetries (i.e. 0.02, 0.05, 0.1 and 0.2; see electronic supplementary material, figures E5.1 – E5.2). In the main text, we depict only autocorrelations characterized by an asymmetry of 0.1. This value can reveal the qualitative differences between symmetric and asymmetric transition probabilities, while still covering a large range of autocorrelations. Specifically, we set $P(E_0 | E_1) - P(E_1 | E_0) = 0.1$, so $E_0$ is the more likely environmental state. For both symmetric and asymmetric cases, we present results for values of $P(E_0 | E_1)$ and $P(E_1 | E_0)$ that approximate autocorrelations of 0.2, 0.5 and 0.8.

## (b) Quantifying plasticity

We simulate experimental designs resembling empirical adoption studies. These studies compare mature organisms, often twins or siblings, separated at a particular point during ontogeny for a specific duration. Researchers investigate how the age at which organisms are separated (and possibly later reunited), and the conditions during separation, determine variation in mature phenotypes. We have previously shown that different manipulations of experiences during separation—for instance, receiving reciprocal opposite cues or cues from a different patch, and temporary or permanent separations—yield similar qualitative patterns [32]. These patterns are most pronounced for reciprocal opposite cues and permanent separation, as experience is maximally divergent for a longer time. Therefore, we analyse only this manipulation here.

We use the optimal policy to simulate developmental trajectories. The level of plasticity corresponds to the extent to which phenotypic development depends on cues during ontogeny. We compute plasticity for each $t \in \{1, T_{ont}\}$. We start by simulating pairs of clones, following the optimal policy. Organisms start in either environment, $E_0$ or $E_1$. We simulate all possible sequences of cues, resulting in one pair of clones per sequence. Each pair of clones receives a weight according to the likelihood of its particular cue sequence.

Clones develop together until time period $t$, experiencing the same sequence of cues and making the same phenotypic decisions, resulting in identical phenotypes. At this point, the clones are separated, with one (the focal) remaining in the original patch and the other (the copy) developing in a mirror patch. The sequence of environmental states in the mirror patch is the same as in the original patch. However, the cues in the mirror patch are opposite to those in the original patch. Whenever the focal individual samples $C_0$, the copy samples $C_1$, and *vice versa*. Focal-and-copy pairs continue development until maturation. Mature phenotypes are described by the number of time periods specialized towards $P_0$ and $P_1$. Together with the number of time periods waited, these numbers sum to $T_{ont} = 10$.

At the end of ontogeny, we compute the weighted average Euclidean distance between the two-dimensional phenotype vectors across all simulated pairs of clones. To control for the number of time periods the focal and copy have developed together, we normalize this measure by dividing the weighted average by the maximally attainable Euclidean distance, resulting in a range from 0 to 1. We show a schematic overview of our adoption study paradigm in the electronic supplementary material, S6, figure E6.1).

## 4. Results

First, we present the optimal phenotypic decisions across ontogeny. Next, we present the levels of plasticity resulting from these policies. We present the linear reward and linear penalty combinations (penalty weight of 1) below and all other combinations in the electronic supplementary material, S7, figures E7.1–E7.54. Additionally, we show distributions of mature phenotypes and compare the terminal fitness of the optimal policies against two non-plastic strategies in the electronic supplementary material, S7.

## (a) Optimal decisions across ontogeny

All organisms start out with the same estimate of the environmental state. These estimates diverge across ontogeny based on individual variation in sampled cues and then converge in adulthood towards the stationary distribution after learning stops (electronic supplementary material, figure E6.2). While adult organisms no longer sample cues, their estimates of the environmental state continue to change, converging towards the stationary distribution. If adulthood is long enough, the estimates across individuals fully converge (electronic supplementary material, figure E6.2).

With symmetric transition probabilities, the optimal developmental decision is to specialize towards the environment with the higher posterior in every time period (figure 1; electronic supplementary material, E7.10). This result follows from two facts. First, the stationary distribution implies that, on average, organisms will encounter each environmental state equally often. Second, organisms never change their estimates about which state is more likely during adulthood (electronic supplementary material, figure E6.2). This means that, if an organism estimates that $E_0$ is the more likely state at the onset of adulthood, it will continue to do so regardless of the duration of adulthood. Taken together, organisms should specialize according to their posteriors regardless of the adult lifespan.

With asymmetric transition probabilities, the optimal decision depends on the relative length of adulthood, the cue reliability and the autocorrelation between environmental states (figure 1; electronic supplementary material, E7.19). When adulthood is long relative to ontogeny (20 time periods) or the cue reliability is low (0.55), organisms always specialize towards $E_0$, the more likely state in the stationary distribution. With a long adult lifespan, organisms will more often encounter $E_0$ during adulthood and specialize accordingly. When cue reliability is poor, organisms remain uncertain about the environmental state when entering adulthood and so choose the more likely state (figure 1). When adulthood is short relative to ontogeny (1 or 5 time periods), there is a high probability that the adult environment differs from the most likely state in the

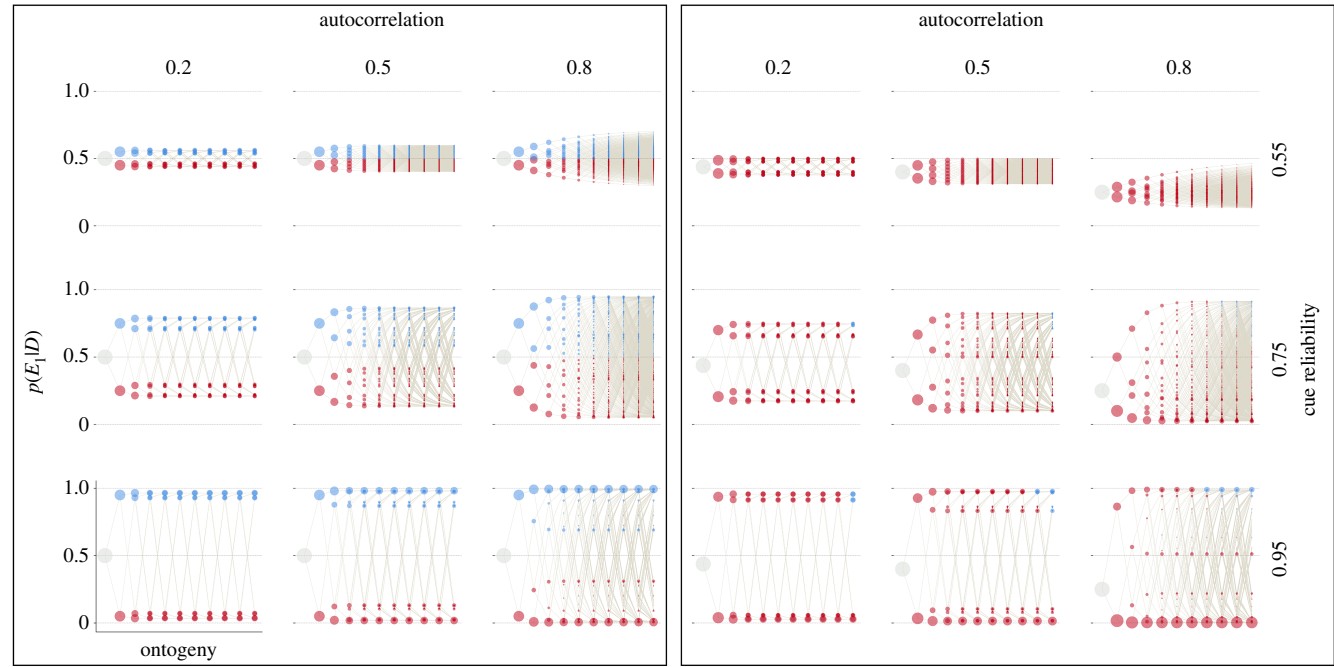

**Figure 1.** Optimal policies. Optimal policies are shown for linear rewards and linear penalties (penalty weight of 1), $T_{adult} = 5$, and symmetric (*left panel*) and asymmetric (*right panel*) transition probabilities. Within each panel, columns indicate different levels of autocorrelation and rows indicate different cue reliabilities. Each combination of asymmetry, autocorrelation level and cue reliability results in a unique Markov process. The vertical axis displays posterior estimates of being in $E_1$ and the horizontal axis displays time during ontogeny. At the onset of ontogeny, all organisms start with a prior estimate of being in $E_1$ according to the stationary distribution (large grey circles). Throughout ontogeny, organisms sample cues and update their posteriors, resulting in the coloured circles. Colours indicate the optimal, fitness-maximizing phenotypic decision in each state. Pies highlight cases in which organisms with the same posterior estimates (but different phenotypic states) make different phenotypic decisions. Black corresponds to waiting (not visible here because organisms never choose to wait), blue to specializing towards $P_1$, red to specializing towards $P_0$. The area of a circle (or pie piece) is proportional to the probability of reaching each state. In each time period, these probabilities sum to 1. Beige lines between states depict possible developmental trajectories. (Online version in colour.)

stationary distribution (electronic supplementary material, figure E6.2). That is, a substantial proportion of organisms—though never the majority—spends more time in $E_1$. The autocorrelation and cue reliability determine when during ontogeny, and with which posteriors, organisms start specializing towards the less likely state. The higher the autocorrelation, the sooner organisms start specializing towards $E_1$ because they can better anticipate the adult environment. With reliable cues (0.75 and 0.95), organisms achieve more extreme posterior estimates and are more likely to specialize based on their posteriors at the end of ontogeny (figure 1; electronic supplementary material, E7.19).

### (b) Optimal levels of plasticity across ontogeny

Fixed, non-plastic policies are favoured only under a narrow range of conditions. When plasticity is favoured, it is retained until the end of ontogeny, though the timing of peak-plasticity varies. With low autocorrelation, plasticity peaks towards the end of ontogeny. With high autocorrelation, the timing of sensitive periods depends upon the cue reliability, with plasticity peaking at the onset, halfway through or towards the end of ontogeny. We elaborate below.

#### (i) Plasticity is not favoured when one environment is more likely and adulthood is long or cues are unreliable

All else equal, asymmetric transition probabilities reduce the scope for plasticity. After all, with one state more likely than

the other, the organism faces less uncertainty and will rely less on plasticity and more on its prior. Asymmetric transition probabilities coupled together with long relative adulthoods result in low plasticity, or even no plasticity, across ontogeny (figure 2). Longer adult lifespans allow organisms to rely on the stationary distribution to adjust to their adult environment, reducing the need for plasticity. The stationary distribution implies that, on average, organisms will encounter the more likely environmental state more often than the less likely state (electronic supplementary material, figure E6.2). Asymmetric transition probabilities coupled together with unreliable cues (0.55) favour zero plasticity across ontogeny (figure 2). To avoid phenotype–environment mismatches due to unreliable cues, organisms use their priors at the onset of ontogeny to specialize towards the more likely environmental state. With symmetric transition probabilities, shorter adult lifespans or reliable cues, plasticity is favoured across ontogeny even when the environment fluctuates frequently (i.e. autocorrelation equals 0.2).

#### (ii) Environmental fluctuations favour sensitive but not critical periods

Unlike previous models that assume stable environments [11,12,32], we find that 'critical periods' in which plasticity drops to zero are never favoured. When the environment fluctuates, organisms always benefit from adjusting their phenotypes—even late into ontogeny. The exact level of

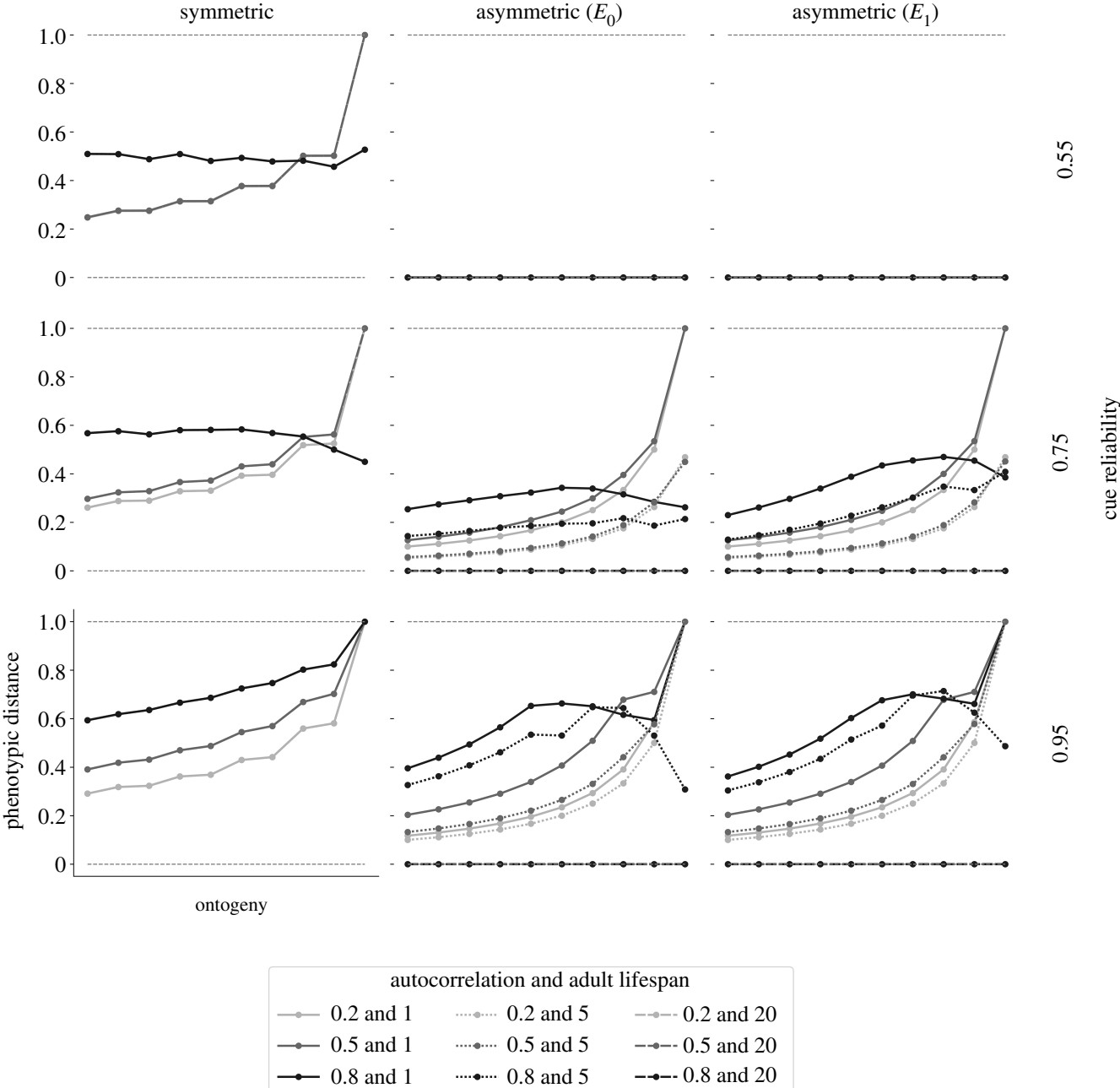

**Figure 2.** Phenotypic plasticity across ontogeny. Phenotypic plasticity across ontogeny is shown for linear rewards and linear penalties (penalty weight of 1). Columns indicate whether transition probabilities are symmetric or asymmetric and in the latter case, whether organisms start development in the more ($E_0$) or less likely ($E_1$) environmental state. Rows indicate different cue reliabilities. Within each panel, we show separate lines for different levels of autocorrelation (indicated by the greyscale) and different adult lifespans (indicated by the line type). Each combination of asymmetry, autocorrelation level and cue reliability results in a unique Markov process. For each combination of a unique Markov process, starting environment and adult lifespan, we conduct $T_{ont} = 10$ experimental twin studies, one for each $t \in \{1, T_{ont}\}$. We simulate $2^{10}$ pairs of clones (one for each possible sequence of cues), who follow the optimal policy and get separated at time period $t$ during ontogeny (horizontal axis). We compute phenotypic distance (vertical axis) as the average, weighted Euclidean distance of all pairs of clones at the end of ontogeny and plot it against the time of separation. Phenotypic distance is normalized by dividing it by the maximally attainable Euclidean distance.

plasticity at the end of ontogeny depends on the adult lifespan and the autocorrelation.

Short adult lifespans (1 time period) favour higher levels of plasticity at the end of ontogeny compared to longer adult lifespans (5 or 20 time periods) (figure 2). When adulthood is short, organisms rely on the most recent cues prior to the onset of adulthood. When adulthood is moderately long (5 time periods), organisms rely on a combination of recent cues and the prior (electronic supplementary material,

figures E7.10 and E7.19). Only those organisms with highly certain posterior estimates specialize towards the less likely state. Those with less certainty specialize towards the more likely state in the stationary distribution. When the adult lifespan is long, natural selection favours non-plastic strategies.

Lower autocorrelations typically result in higher levels of plasticity at the end of ontogeny (figure 2). The more frequent environmental fluctuations are, the more cues can shift

posterior estimates throughout all of ontogeny, increasing the scope for plasticity (figure 1). When the autocorrelation is high (0.8), organisms are less likely to attend to cues towards the end of ontogeny, resulting in lower levels of end-of-ontogeny plasticity. A relatively stable environment allows them to reduce uncertainty about their adult environment earlier during ontogeny. However, when cues are highly reliable (0.95) and the adult lifespan is short (1 time period), the chance of sampling incorrect cues is so low that the expected benefits from additional information about the environment outweigh potential mismatch costs. Under these conditions, end-of-ontogeny levels of plasticity can match those of ecologies with lower autocorrelations.

### (c) The timing of sensitive periods across ontogeny

#### (i) Sensitive periods can evolve at the onset of ontogeny when environmental fluctuations are rare (autocorrelation 0.8)

With symmetric transition probabilities and moderately reliable cues (0.75), organisms initially use cues to reduce uncertainty about their environment, resulting in a constant level of plasticity over large portions of ontogeny (figure 2). However, plasticity declines towards the end as some organisms achieve more extreme posterior estimates and consistently specialize towards one phenotypic target (figure 1). When cue reliability is low (0.55), natural selection favours constant, non-zero levels of plasticity across ontogeny. Neither the stationary distribution nor the sampled cues provides sufficient information to reduce uncertainty about the state of the environment. Organisms remain fairly uncertain and thus attend to noisy cues across all of ontogeny.

#### (ii) Sensitive periods may evolve halfway through ontogeny when environmental fluctuations are rare (autocorrelation 0.8)

With asymmetric transition probabilities and reliable cues (0.75 or 0.95), organisms specialize early in ontogeny according to the stationary distribution, ignoring sampled cues. As ontogeny proceeds, plasticity increases because organisms become more uncertain when they sample cues that contradict their posteriors. Plasticity peaks when organisms reach states after which they are likely to consistently specialize towards one phenotypic target, reducing the scope for plasticity in subsequent time periods. When cues are moderately reliable (0.75) and the adult lifespan is short (1 time period), organisms may reach such states halfway through ontogeny. Highly reliable cues (0.95) increase the probability that organisms start to specialize towards the less likely state already halfway through ontogeny (figure 2), for both short and moderate adult lifespans (1 or 5 time periods).

#### (iii) Sensitive periods often evolve towards the end of ontogeny

Frequent environmental fluctuations favour sensitive periods towards the end of ontogeny. In such conditions (autocorrelations of 0.2 and 0.5), organisms specialize according to the most recent cues prior to the onset of adulthood (figures 1 and 2). When environmental fluctuations are rare (autocorrelation of 0.8), plasticity sometimes peaks towards the end of ontogeny. When the adult lifespan is moderate (5 time periods) and cues are moderately reliable (0.75), a small proportion of the population specializes towards the less likely state in later time periods, resulting in sensitive periods

towards the end of ontogeny. Plasticity may also peak at the end of ontogeny when the adult lifespan is short (1 time period) and cues are highly reliable (0.95), because organisms always choose to specialize according to cues in the final time period (figures 1 and 2). These are also the only conditions in our model that favour two peaks in plasticity: one smaller peak halfway through ontogeny and one larger peak in the final time period. In the second half of ontogeny, plasticity decreases because many organisms are locked into developmental trajectories on which they consistently specialize towards the same state. However, to reduce mismatch penalties during a short adulthood, organisms always specialize according to the final cue as a form of insurance.

## 5. Discussion

### (a) Sensitive periods are more likely to evolve than critical periods

Unlike in models assuming stable environments, we find that critical periods, in which plasticity drops to zero, are never favoured. In a fluctuating environment, organisms always use cues at the end of ontogeny to reduce uncertainty about their adult environment. Combining insights across models, we may expect empirical researchers to observe critical periods for traits that have evolved in stable ontogenetic environments and sensitive periods for traits that have evolved in fluctuating ontogenetic environments.

Our finding that plasticity always persists at the end of ontogeny is striking for two reasons. First, previous work shows that the fitness costs of plasticity may outweigh the fitness benefits when organisms need to continuously re-adjust to fluctuating environmental conditions and pay the associated costs [4,34,35]. Our model assumes no costs to building, maintaining and running the physiological machinery for plasticity, to sampling cues and to making phenotypic adjustments. Our model does, however, assume that plasticity is incremental and irreversible, and that there is a cost to phenotype–environment mismatch in adulthood. With these assumptions, the level of plasticity at the end of ontogeny is highest when adulthood is short and the rate of environmental fluctuations is high. Second, previous models exploring the evolution of plasticity in fluctuating environments have assumed that developed phenotypes can be undone, allowing organisms to continuously readjust their phenotypes to changing conditions [34,36]. The ability to reverse development may reduce phenotype–environment mismatch and thus make plasticity across all of ontogeny more viable. In our model, organisms cannot reverse phenotypic increments. Developmental trajectories, however, are reversible, such that organisms may specialize towards the opposite phenotypic target at any point during ontogeny. This allows phenotypic plasticity to be favoured and even persist until the end of ontogeny when the environment fluctuates frequently.

A study by Relyea [37] has shown that tadpoles (*Hyla versicolor*)—which cannot undo developed phenotypes but are able to switch developmental trajectories—retain plasticity towards the end of ontogeny in a fluctuating environment. Tadpoles were exposed to variation in predation risk across ontogeny and showed the induction of morphological defenses, such as greater mass, deeper tails

or shorter bodies, throughout all of ontogeny, albeit to a lesser extent at later stages. Relyea did not operationalize reversibility as the deconstruction of phenotypic adjustments, but as the relative readjustment of different morphological features. For example, if a tadpole developed a deeper tail relative to its body size in response to predators and increased body size after predators were later removed, this counted as a reversal. Such reversals are similar to the reversibility of specialization trajectories in our model. The study showed that reversibility of phenotypic inductions was high early in ontogeny and lower later during ontogeny. Our model predicts a decline in organisms' ability to switch between trajectories as ontogeny progresses, when environmental fluctuations are rare and cues are moderately reliable. In such conditions, the remaining time is too short for organisms to revise estimates. Relyea allowed for a switch in predation risk only once during ontogeny. These conditions resemble those of high autocorrelation values in our model, more so than those of low autocorrelation. However, we do not know the reliability of predation cues used in the study. Future research could replicate the experiment while manipulating the reliability of cues. Our model predicts a steep increase in plasticity at the end of ontogeny when cues are highly reliable.

## (b) The timing of sensitive periods

We find that sensitive periods typically occur at the end of ontogeny. This finding contrasts with results from models of stable environments [10–15], as well as the results of one model exploring fluctuating environments [16]. Our finding indicates that natural selection may heighten sensitivity to cues towards the end of ontogeny when the environment changes rapidly and phenotypes develop incrementally. Developing organisms then use experiences towards the end of ontogeny to adjust phenotypes right before maturity. This makes sense: when the environment fluctuates frequently, cues towards the end of ontogeny tend to better predict conditions in adulthood than earlier cues. Responses to cues can be behavioural or morphological. Examples of greater reliance on cues later during development exist for both. For example, migratory bird, bat and fish species use cues throughout their journeys to predict remote conditions and adjust their arrival time and destination [38]. Often, these animals rely the most on cues towards the end of their journey to make such predictions. In bulb mites (*Rhizoglyphus robini*), nutritional conditions during the final ontogenetic stages determine whether males mature as 'fighters' or 'scramblers' [39]. The extent to which these patterns depend on the rate of environmental change, the reliability of cues or the duration of adulthood relative to ontogeny remain to be explored. Experimental evolution studies of bulb mites and other insect species can be used to explore different parameter combinations and test predictions from models like ours [40].

While sensitive periods often emerge at the end of ontogeny in a fluctuating environment, they may occur midway through ontogeny when autocorrelation is high. Models of stable environments have obtained this same result [15,32], and so did Fischer *et al.*'s [16] model of fluctuating environments [16]. These sets of models produce this pattern, at least in part, for the same reason: the initial discrepancy between posteriors and cues increases uncertainty, which increases plasticity early in ontogeny, before plasticity

declines when later cues reduce uncertainty. Our model also produces sensitive periods midway through ontogeny, but for different reasons from those models. Fischer *et al.* [16] and Stamps & Krishnan [15] assume that organisms start development with specialized phenotypes and that specializations are fully reversible. Under these conditions, sensitive periods may arise midway through ontogeny because organisms first sample multiple cues to reduce uncertainty, before changing their specialized phenotypes. This effect may be strongest when phenotypic adjustments are costly, as in Fischer *et al.*'s [16] model. Complete reversibility further allows organisms to delay developing specializations because the scope for phenotypic adjustment is not constrained by the duration of ontogeny. In the current model, and in a previous model of stable environments with variation in the cue reliability across ontogeny [32], sensitive periods midway through ontogeny are favoured even if organisms do not start out with specialized phenotypes that are costly to switch away from and adjustments are irreversible. Thus, sensitive periods midway through ontogeny may evolve across a range of environments and life histories.

## (c) Long adult lifespans disfavour plasticity

When adulthood is short relative to ontogeny, high levels of plasticity are favoured across ontogeny. By contrast, when adulthood is long, organisms rely less on (or ignore) their experiences and specialize towards the more likely state in the stationary distribution. These findings may appear at odds with those of a coevolutionary model by Ratikainen & Kokko [41] showing that longevity favours plasticity and *vice versa*. However, this difference can be understood in light of assumptions about plasticity in adulthood. Our model assumes that phenotypic development is limited to ontogeny. Their model allows adults to continue tailoring their phenotypes. Thus, in their model (but not in ours), long-lived organisms can do better than adapting to the stationary distribution. Combining both models, we may predict that longevity is associated with higher levels of plasticity when adult phenotypes are malleable, and with lower levels of plasticity when adult phenotypes are fixed. Cross-species comparisons have shown that higher levels of plasticity are associated with longer lifespans in some groups of animals [42,43], but with shorter lifespans in other groups of animals [44]. Future work may test whether the malleability of adult phenotypes moderates these opposite patterns of association.

## (d) Limitations and future directions

Our model assumes only two environmental states. Another possibility would be to assume a larger number of discrete states or a continuum of states. This would make it possible to independently manipulate the means and variances of both the prior distribution and the reliability of cues. Doing so might influence the findings from our model. However, previous models of stable environments that incorporate a continuum of environmental states [13,15] have found similar qualitative patterns to those assuming two discrete states [11,12]. Future modelling could explore whether our results replicate when increasing the number environmental states.

In our model, fitness is only a function of fertility. Other state-dependent models assume that fitness depends on fertility and mortality [16,45]. Our model could be extended to

include mortality. Mortality would be a function of phenotype–environment match during ontogeny, adulthood, or both, depending on how the trait influences mortality across these stages.

In our model, fitness is proportional to the difference between correct and incorrect specializations. We instantiate this through specific reward–penalty mappings and penalty weights. In the main text, we set the penalty weight to 1, implying that rewards and penalties contribute equally to fitness. In the electronic supplementary material, we show that penalty weights of 0.5 and 2 yield the same qualitative patterns, but there is one difference: when the penalty weight is 2, organisms sometimes wait to reduce phenotype–environment mismatch. Though we have explored a wide parameter range, future work could investigate a more general model where fitness is an arbitrary function of phenotype.

Our model assumes that organisms 'know' (i.e. have evolutionarily adapted to) the cue reliability, autocorrelation, and the durations of ontogeny and adulthood, because these parameters were fixed across generations. However, if these parameters were variable, organisms may estimate them based on experience. Future modelling could explore a scenario in which the cue reliability, autocorrelation and the duration of life stages vary between generations, but are stable within generations. For instance, an organism might be born into one of several patches, each of which has its own cue reliability, autocorrelation or duration of life stages. Future modelling could also explore a scenario in which these parameters vary within generations as well. For instance, the weather might change at different rates in different seasons. Under these conditions, organisms may need to learn the pattern of change of environmental parameters across their lifespan [46]. In experimental studies, humans, non-human primates and rodents are able to learn the cue reliability and adaptively adjust their behaviour [47,48]. It would be interesting to see whether organisms that are uncertain about multiple parameters retain higher end-of-ontogeny levels of plasticity, as we see in the current model. Organisms may develop sensitive periods late in ontogeny, if conditions favour attaining confident estimates of environmental parameters prior to committing to phenotypic specialization.

As noted, animals and plants may experience fluctuations in different environmental statistics during their lifetimes [49]. For example, the reliability of cues varies across ontogeny for a variety of aquatic species, such as larval mosquitos (*Culex restuans*), common roaches (*Rutilus rutilus*), fathead minnows (*Pimephales promelas*) and goldfish (*Carassius auratus*) [50]. However, for many species and traits, there is little information about the values of environmental statistics across ontogeny [51]. As a future direction, we envision a repository of environmental statistics across ontogeny (e.g. autocorrelation, cue reliability) for a range of species and populations. Such a repository can benefit empirical researchers who study how environmental conditions shape development, as well as theoreticians modelling the evolution of developmental phenomena, such as sensitive and critical periods. For instance, it would allow modellers to make informed decisions about which parameters to fix or vary across ontogeny, depending on their research questions about groups of organisms (e.g. taxa, clades) or particular species or populations. Modellers and empirical researchers may use the repository to focus on those rates of variation that are most relevant for a given taxonomic group or species when developing theory and experiments. In this way, a repository of environmental statistics has the potential to strengthen connections and create synergies between empirical and theoretical work, thus accelerating progress in our understanding of the evolution and development of sensitive periods.

**Ethics.** The study was conducted according to the Declaration of Helsinki and was approved by DTU Compute's Institutional Review Board (COMP-IRB-2020-02).

**Data accessibility.** The (preprocessed) data analysed in this manuscript and the code used for data analysis are available on Github (see https://github.com/Nicole-Walasek/SensitivePeriodsInFluctuatingEnvironments).

The data are provided in the electronic supplementary material [52].

**Authors' contributions.** N.W.: conceptualization, formal analysis, methodology, visualization, writing—original draft and writing—review and editing; W.E.F.: conceptualization, funding acquisition, supervision, writing—original draft and writing—review and editing; K.P.: conceptualization, supervision, writing—review and editing.

All authors gave final approval for publication and agreed to be held accountable for the work performed therein.

**Competing interests.** We declare we have no competing interests.

**Funding.** We received funding for this study from Jacobs Foundation (grant no. 2017 1261 02), James S. McDonnell Foundation (grant no. https://doi.org/10.37717/220020502) and Nederlandse Organisatie voor Wetenschappelijk Onderzoek (grant no. V1.Vidi.195.130).

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
