## [Peer Review File · Proceedings of the Royal Society B: Biological Sciences]

Review History

RSPB-2021-1692.R0 (Original submission)

Review form: Reviewer 1

Recommendation

Major revision is needed (please make suggestions in comments)

Scientific importance: Is the manuscript an original and important contribution to its field?

Good

General interest: Is the paper of sufficient general interest?

Good

Quality of the paper: Is the overall quality of the paper suitable?

Poor

Is the length of the paper justified?

Yes

Should the paper be seen by a specialist statistical reviewer?

No

Do you have any concerns about statistical analyses in this paper? If so, please specify them explicitly in your report.

No

It is a condition of publication that authors make their supporting data, code and materials available - either as supplementary material or hosted in an external repository. Please rate, if applicable, the supporting data on the following criteria.

Is it accessible?

Yes

Is it clear?

Yes

Is it adequate?

Yes

Do you have any ethical concerns with this paper?

No

Comments to the Author

See attached file. (See Appendix A)

Review form: Reviewer 2

Recommendation

Major revision is needed (please make suggestions in comments)

Scientific importance: Is the manuscript an original and important contribution to its field?

Excellent

General interest: Is the paper of sufficient general interest?

Good

Quality of the paper: Is the overall quality of the paper suitable?

Excellent

Is the length of the paper justified?

Yes

Should the paper be seen by a specialist statistical reviewer?

No

Do you have any concerns about statistical analyses in this paper? If so, please specify them explicitly in your report.

No

It is a condition of publication that authors make their supporting data, code and materials available - either as supplementary material or hosted in an external repository. Please rate, if applicable, the supporting data on the following criteria.

Is it accessible?

N/A

Is it clear?

N/A

Is it adequate?

N/A

Do you have any ethical concerns with this paper?

No

Comments to the Author

The manuscript describes a model that is used to investigate which conditions favour critical or sensitive periods during development for plastic adjustments to the phenotype. Overall, I think the model results are interesting and the manuscript is well written. I therefore have only a few comments and suggestions for the authors:

l. 45-47: It is a bit difficult to understand what critical periods in this sentence alone. I also find this sentence to not be quite necessary and would suggest you cut this. If not, please rephrase.

l. 71/75: Again, the definition of "critical periods" is not quite clear. In order to accommodate a wider audience, I think it is necessary to explain this a bit further before this term is used as here.

l.113 and after: You state that phenotypic change is irreversible, but the way I understand your model an individual that has specialized once towards each environmental state is back to start. If this is correct then the model would be equivalent to one where phenotypic change is reversible. This is mentioned several times and is an important point in the discussion as well, so some further clarification is strongly encouraged.

l. 228/239-240: The point of, and therefore the meaning of "t" is unclear on line 228, maybe move the info on l 139-240 up?

236-240: Your measure of stage-specific plasticity is very important for the understanding of the results, and seems appropriate to me, but it took me a long time to understand what this measure is and how it is calculated, so I think it is necessary with a better explanation of this. A figure might be helpful here.

Figure 1:

Y-axis label, left side ("estimate") is not very explanatory, could you relabel?

L 279-280. I presume this is the estimate of the probability of being in E1?

l. 281/282: please specify what the prior and posterior refer to.

l. 284: I can't see black appear in this graph?

Figure 2:

This figure is possibly more confusing than helpful. You seem to need more dimensions than available and therefore have adult lifespan nested within transition probability and autocorrelation nested within cue reliability. While this might be necessary to show all results in one figure, it needs to be much clearer that the two major axes are not at all continuous.

Since the axes are not continuous, the areas with black frames does not make complete sense to me. There is not reason why asymmetric E0 should be left of asymmetric E1, and the picture would look quite different if these switched places. It would also be possible to nest e.g. transition probabilities within adult lifespan, which would also completely change the picture. A figure that makes the nesting more obvious might be better, but it is not quite clear to me what the goal of this figure is.

Figure 3:

It is again a bit difficult to understand the measure on the y-axis and the label is not quite self-explanatory. Right now it looks like all phenotypes are always very similar at the end of ontogeny.

Decision letter (RSPB-2021-1692.R0)

06-Sep-2021

Dear Ms Walasek:

I am writing to inform you that we have now obtained responses from referees on manuscript RSPB-2021-1692 entitled "Sensitive periods, but not critical periods, evolve in a fluctuating environment: A model of incremental development" which you submitted to Proceedings B.

Unfortunately, on the advice of the Associate Editor and the referees, your manuscript has been rejected following full peer review. Competition for space in Proceedings B is currently extremely severe, as many more manuscripts are submitted to us than we have space to print. We are therefore only able to publish those that are exceptional, convincing and present significant advances of broad interest, and must reject many good manuscripts.

Please find below the comments received from referees concerning your manuscript, not including confidential reports to the Editor. I hope you may find these useful should you wish to submit your manuscript elsewhere.

We are sorry that your manuscript has had an unfavourable outcome, but would like to thank you for offering your work to Proceedings B.

Sincerely,
Dr Sasha Dall
mailto:proceedingsb@royalsociety.org

Associate Editor
Board Member: 1
Comments to Author:

This manuscript addresses an interesting and important aspect of phenotypic plasticity. If the environment changes throughout development, and if organisms can alter their developmental trajectory continually in response to cues that are associated with these changes, what is the optimal developmental strategy? Unfortunately though, I think that R1 has identified a critical problem with the model. Undoubtedly the order in which the environmental states occur will affect the optimal developmental strategy but the model does not account for this. Instead, it assumes that all that matters is the number of times the environment is in each state during development but not the order in which they occur. I think this excludes a fundamentally important feature of the problem and so, unless we have both missed something, a different model would be required to analyze this question in a more satisfying way.

Reviewer(s)' Comments to Author:
Referee: 1
Comments to the Author(s)
See attached file

Referee: 2
Comments to the Author(s)

The manuscript describes a model that is used to investigate which conditions favour critical or sensitive periods during development for plastic adjustments to the phenotype. Overall, I think the model results are interesting and the manuscript is well written. I therefore have only a few comments and suggestions for the authors:

1. 45-47: It is a bit difficult to understand what critical periods in this sentence alone. I also find this sentence to no be quite necessary and would suggest you cut this. If not, please rephrase.

l. 71/75: Again, the definition of “critical periods” is not quite clear. In order to accommodate a wider audience, I think it is necessary to explain this a bit further before this term is used as here.

l.113 and after: You state that phenotypic change is irreversible, but the way I understand your model an individual that has specialized once towards each environmental state is back to start. If this is correct then the model would be equivalent to one where phenotypic change is reversible. This is mentioned several times and is an important point in the discussion as well, so some further clarification is strongly encouraged.

l. 228/239-240: The point of, and therefore the meaning of “t” is unclear on line 228, maybe move the info on l 139-240 up?

236-240: Your measure of stage-specific plasticity is very important for the understanding of the results, and seems appropriate to me, but it took me a long time to understand what this measure is and how it is calculated, so I think it is necessary with a better explanation of this. A figure might be helpful here.

Figure1:

Y-axis label, left side (“estimate”) is not very explanatory, could you relabel?

L 279-280. I presume this is the estimate of the probability of being in E1?

l. 281/282: please specify what the prior and posterior refer to.

l. 284: I can’t see black appear in this graph?

Figure 2:

This figure is possibly more confusing than helpful. You seem to need more dimensions than available and therefore have adult lifespan nested within transition probability and autocorrelation nested within cue reliability. While this might be necessary to show all results in one figure, it needs to be much clearer that the two major axes are not at all continuous. Since the axes are not continuous, the areas with black frames does not make complete sense to me. There is not reason why asymmetric E0 should be left of asymmetric E1, and the picture would look quite different if these switched places. It would also be possible to nest e.g. transition probabilities within adult lifespan, which would also completely change the picture. A figure that makes the nesting more obvious might be better, but it is not quite clear to me what the goal of this figure is.

Figure 3:

It is again a bit difficult to understand the measure on the y-axis and the label is not quite self-explanatory. Right now it looks like all phenotypes are always very similar at the end of ontogeny.

Author's Response to Decision Letter for (RSPB-2021-1692.R0)

See Appendix B.

RSPB-2021-1692.R1 (Revision)

Review form: Reviewer 1

Recommendation

Major revision is needed (please make suggestions in comments)

Scientific importance: Is the manuscript an original and important contribution to its field?

Good

General interest: Is the paper of sufficient general interest?

Good

Quality of the paper: Is the overall quality of the paper suitable?

Good

Is the length of the paper justified?

Yes

Should the paper be seen by a specialist statistical reviewer?

No

Do you have any concerns about statistical analyses in this paper? If so, please specify them explicitly in your report.

No

It is a condition of publication that authors make their supporting data, code and materials available - either as supplementary material or hosted in an external repository. Please rate, if applicable, the supporting data on the following criteria.

Is it accessible?

N/A

Is it clear?

N/A

Is it adequate?

N/A

Do you have any ethical concerns with this paper?

No

Comments to the Author

See attached file. (See Appendix C)

Decision letter (RSPB-2021-2623.R0)

04-Jan-2022

Dear Ms Walasek:

Your manuscript has now been peer reviewed and the reviews have been assessed by an Associate Editor. The reviewers' comments (not including confidential comments to the Editor) and the comments from the Associate Editor are included at the end of this email for your reference. As you will see, the reviewers and the Editors have raised some concerns with your manuscript and we would like to invite you to revise your manuscript to address them.

We do not allow multiple rounds of revision so we urge you to make every effort to fully address all of the comments at this stage. If deemed necessary by the Associate Editor, your manuscript will be sent back to one or more of the original reviewers for assessment. If the original reviewers

are not available we may invite new reviewers. Please note that we cannot guarantee eventual acceptance of your manuscript at this stage.

Research ethics:

Use of animals and field studies:

It is a condition of publication that you make available the data and research materials supporting the results in the article (<https://royalsociety.org/journals/authors/author-guidelines/#data>). Datasets should be deposited in an appropriate publicly available repository and details of the associated accession number, link or DOI to the datasets must be included in the Data Accessibility section of the article (<https://royalsociety.org/journals/ethics-policies/data-sharing-mining/>). Reference(s) to datasets should also be included in the reference list of the article with DOIs (where available).

Please submit a copy of your revised paper within three weeks. If we do not hear from you within this time your manuscript will be rejected. If you are unable to meet this deadline please let us know as soon as possible, as we may be able to grant a short extension.

Best wishes,
Dr Sasha Dall
mailto:proceedingsb@royalsociety.org

Associate Editor Board Member

Comments to Author:

The authors have done a nice job of fixing the main problem with the previous model but there are a couple of important issues remaining that I think they need to address. Perhaps the most important is being clear about potential real-world biological scenarios where the model assumptions would apply. There are also some (hopefully minor) outstanding issues regarding the mathematical presentation.

Reviewer(s)' Comments to Author:

Referee: 1

Comments to the Author(s).

See attached file.

Author's Response to Decision Letter for (RSPB-2021-2623.R0)

See Appendix D.

Decision letter (RSPB-2021-2623.R1)

24-Jan-2022

Dear Nicole

I am pleased to inform you that your manuscript entitled "Sensitive periods, but not critical periods, evolve in a fluctuating environment: A model of incremental development" has been accepted for publication in Proceedings B.

Data Accessibility section

Open Access

Paper charges

Sincerely,

Dr Sasha Dall

Appendix A

Comments on Walasek et al.

This ms provides a simple model that attempts to understand developmental plasticity when changes during development are incremental. The effects of incremental change are worth exploring as nature works in that way. Of course the model has no costs of change, and so is rather limited. Nevertheless, one has to start somewhere, so I am not suggesting that costs should be introduced. From the current ms, I do not have a sense of how allowing changes to occur faster would change results. It would be good to have that intuition.

The ms also varies the time horizon after maturation, which can then be used to explore how the likelihood of change after maturation effects results. This is an interesting issue. I am not sure that varying the time horizon is the best way to get at this question, but it is one way.

Overall the general approach of the ms has the potential to make a worthwhile contribution to the literature. However, there appears to be one serious problem with the analysis (see directly below) and various other issues that require attention.

A problem with the state representation. In order to calculate the optimal strategy under dynamic programming the model takes as its informational state variable $D = (c_0, c_1)$, where c_0 and c_1 are the number of times each cue has been received so far. If I have understood the analysis in the paper correctly, the order of the cues is not important. Thus, say, receiving 5 C_0 followed by 5 C_1 is taken to be equivalent to receiving 5 C_1 followed by 5 C_0 . However, these two sequences lead to different posterior probabilities at time $T = 10$. This is because the environment is changing, implying that more weight should be placed on more recent cues. I may have misinterpreted the model, but I do not think so. For example

under my interpretation there should be 11 possible D values at final time $T = 10$, and this appears to be the case in the figures. If instead, the order is taken into account then we would expect $2^{10} = 1024$ possible states at time $T = 10$. Ignoring the order will completely distort results. So if I am correct, I am afraid the whole model needs to be changed. To model the situation taking the order into account one could take the ordered sequence of previous cues as the current informational state. Alternatively one could just take the Bayes posterior as the state variable. This latter approach is certainly conceptually simpler. The drawback is that the posterior lies on a continuum, so that a fine grid with grid interpolation is necessary in order to do dynamic programming. Nevertheless, one should be able to dynamic program over many time steps relatively quickly.

Other major points

- The motivation for the two states is that one might be safe and the other dangerous (lines 144-145). Furthermore, induced defences (as in *Daphnia*) provide the motivation for specialisation. Strictly speaking, the current model is not applicable in this case as the cost of having low induced defences is that the organism expects a shorter life. However, there is no mortality in the model and individuals all have exactly the same lifespan regardless of their strategy and the environment. I am not suggesting altering the model to include mortality. However, the authors cannot just use the induce defence example as motivation without a ‘health warning’.
- There is insufficient motivation of the payoff structure, even in the case of a linear payoffs. Consider the linear case. In this case the terminal reward at maturation is a linear function of $y_1 - y_0$ (equation 11 below). This fact is not brought out clearly. In this case one can try to give a motivation in terms of the induced defence example: y_1 is investment in defence, y_0 is removing existing investment. So the final $y_1 - y_0$ is the defence at maturity. Is this reasonable

- maybe. The important point is that the ms does not say enough about the biological interpretation of the reward structure, even in this linear case. Note that in the linear case one can just use $z = y_1 - y_0$ as the physiological state variable in the dynamic programming. This reduces the dimension by 1.

- So what about the motivation for the non-linear case. Now the terminal reward is a function of $f(y_1) - f(y_0)$. How are we to interpret this? For the functions used in the non-linear case the terminal reward is no longer a function of $y_1 - y_0$. So what does this mean? It is like there are two traits that control the vulnerability to predation??
- The autocorrelation was found by simulation, whereas it is easy to calculate. I derive the formula in the technical details below (equation 7). This formula should be in various elementary texts on time series so the authors should just quote it and do not need to derive it. It might be worth quoting this simple formula in the main text in order to give the reader a better intuition.

Minor points

- Lines 124-128. The transition probabilities are over unit time. However, this is not said and the time structure has yet to be introduced.
- Might it be worth plotting the terminal reward?
- It appears that in looking at the sensitivity to cues, the cues in the mirror patch are opposite those in the original patch from a given time onwards. If that is true I personally find the measure rather bizarre. Surely it would be more informative just to flip one cue, keeping cues before and after the same?
- The figure captions are not very informative as to underlying parameter values.

- Lines 283-284. It says that ‘Pies highlight cases in which organisms with the same estimates make different phenotypic decisions’. I am unclear what this means.
- The model has just two environmental states, so that the Bayes posterior is characterised by a single variable so that the posterior variance is a function of the posterior mean. In contrast, if there were a continuum of environmental state, the mean does not determine the variance. One can when have the posterior variance decreasing over time even though the mean stays roughly the same. Prediction will then be altered. It might be worth drawing this to readers’ attention, and perhaps mentioning different biological scenarios that illustrate this.

Technical details which may be useful to the authors

Warning: I have used a different notation to the ms in what follows. That is because I personally find the standard mathematical notation I use clearer. I am not suggesting the authors adopt this notation.

Let $X_t = 0$ if the environmental state at time t is E_0 , with $X_t = 1$ if the environmental state at time t is E_1 . I use the notation that $\mathbb{P}(X_{t+1} = 1|X_t = 0) = \lambda_1$ and $\mathbb{P}(X_{t+1} = 0|X_t = 1) = \lambda_0$.

Posterior distribution changes with no cues.

Let $p(t) = \mathbb{P}(X_t = 1)$. Then

$$p(t+1) = (1 - p(t))\lambda_1 + p(t)(1 - \lambda_0). \quad (1)$$

To find the stationarity probability we set $p(t+1) = p(t) = p^*$ in equation 1 to give

$$p^* = \frac{\lambda_1}{\lambda_0 + \lambda_1}. \quad (2)$$

From equation 1 we can then get

$$p(t+1) - p^* = \rho [p(t) - p^*], \quad (3)$$

where

$$\rho = 1 - (\lambda_0 + \lambda_1) \quad (4)$$

is the autocorrelation (see below). Applying this formula recursively gives

$$p(t) = p^* + \rho^t [p(0) - p^*]. \quad (5)$$

Autocorrelation

Note that at stationarity we have $\mathbb{E}(X_t) = \mathbb{E}(X_{t+1}) = p^*$. We also have $\mathbb{E}(X_t X_{t+1}) = \mathbb{P}(X_t = 1, X_{t+1} = 1) = \mathbb{P}(X_t = 1)\mathbb{P}(X_{t+1} = 1|X_t = 1) = p^*(1 - \lambda_0)$. From these equation we get

$$\text{Cov}(X_t, X_{t+1}) = \mathbb{E}(X_t X_{t+1}) - \mathbb{E}(X_t)\mathbb{E}(X_{t+1}) = p^*(1 - \lambda_0 - p^*). \quad (6)$$

It is also easy to see that $\text{Var}(X_t) = \text{Var}(X_{t+1}) = p^*(1 - p^*)$. Thus the correlation between X_t and X_{t+1} is

$$\rho = \frac{\text{Cov}(X_t, X_{t+1})}{\sqrt{\text{Var}(X_t)\text{Var}(X_{t+1})}} = 1 - (\lambda_0 + \lambda_1). \quad (7)$$

The terminal reward

To simplify notation let $p(t)$ denote the probability that the environment is E_1 after t time steps in the adult phase, conditional on the information available at the end of the juvenile phase: in the authors' notation $p(t_{adult}) = \mathbb{P}(E_{1,t_{adult}}|D)$. Thus $p(0)$ is the posterior at the end of the juvenile phase and $p(t)$ is given by equation 5.

We can write line 339-340 of ESM as

$$\phi(y_0, y_1, t) = (1 - p(t))f(y_0) + p(t)f(y_1) \quad (8)$$

$$\psi(y_0, y_1, t) = - [(1 - p(t))f(y_1) + p(t)f(y_0)]. \quad (9)$$

Thus

$$\phi(y_0, y_1, t) + \psi(y_0, y_1, t) = [f(y_1) - f(y_0)] (2p(t) - 1). \quad (10)$$

It then follows that fitness can be expressed as

$$\pi(y_0, y_1) = \pi_0 + [f(y_1) - f(y_0)] L, \quad (11)$$

where L depends on $\{p(t) : t = 0, 1, \dots, T_{adult}\}$. Equation 5 can then be used to find an analytic expression for L (by just summing a geometric progression). This expression might be a bit messy in general, but is especially simple in the symmetric case.

REVISION

Sensitive periods, but not critical periods, evolve in a fluctuating environment: A model of incremental development

Dear Dr. Dall,

We would like you to consider our revision of – “Sensitive periods, but not critical periods, evolve in a fluctuating environment: A model of incremental development” – for publication in *Proceedings B* (RSPB-2021-1692). We are extremely grateful that we were given the opportunity to resubmit our manuscript after appealing the decision to reject. Following the thoughtful and constructive comments from the editor and reviewers, we have substantially revised our manuscript and model. In brief, we have corrected the flaw detected by R1 (for which we are grateful). Our original findings replicate after the model changes, and we have one additional finding. As a result, the qualitative patterns that inform our conclusions have become clearer and more pronounced. With the reviewers’ suggestions incorporated, this model will advance the field even more than our original submission did. In the letter that follows, we address all of the comments from the editor and reviewers.

Additional information about our submission:

- Our submission includes figures and online enhancements (supplementary materials and code).
- Our submission does not overlap with any other published, in press, or in preparation articles, books, or proceedings. Our research is not under consideration elsewhere.

Sincerely,

Nicole Walasek¹, Willem E. Frankenhuis^{1,2,3} and Karthik Panchanathan⁴

¹ Behavioral Science Institute, Radboud University, the Netherlands

² Department of Psychology, Utrecht University, the Netherlands

³ Max Planck Institute for the Study of Crime, Security and Law, Germany

⁴ Department of Anthropology, University of Missouri, U.S.A.

This research was supported by grants from the Dutch Research Council (V1.Vidi.195.130) to WEF, the James S. McDonnell Foundation (<https://doi.org/10.37717/220020502>) to WEF, and the Jacobs Foundation (2017 1261 02) to WEF.

Correspondence concerning this article should be addressed to Nicole Walasek, Thomas van Aquinostraat 4, 6525 GD, Nijmegen, The Netherlands. Phone: +31629386516. E-mail: nicole.walasek@ru.nl.

Responses to editor and reviewer comments

The original comments are in **black** and our replies are in **blue**. Text inserted from the manuscript is in **green**.

XX

EDITORIAL LETTER

XX

Dear Ms Walasek:

I am writing to inform you that we have now obtained responses from referees on manuscript RSPB-2021-1692 entitled "Sensitive periods, but not critical periods, evolve in a fluctuating environment: A model of incremental development" which you submitted to Proceedings B.

Unfortunately, on the advice of the Associate Editor and the referees, your manuscript has been rejected following full peer review. Competition for space in Proceedings B is currently extremely severe, as many more manuscripts are submitted to us than we have space to print. We are therefore only able to publish those that are exceptional, convincing and present significant advances of broad interest, and must reject many good manuscripts.

Please find below the comments received from referees concerning your manuscript, not including confidential reports to the Editor. I hope you may find these useful should you wish to submit your manuscript elsewhere.

We are sorry that your manuscript has had an unfavourable outcome, but would like to thank you for offering your work to Proceedings B.

Sincerely,

Dr Sasha Dall
mailto: proceedingsb@royalsociety.org

Associate Editor
Board Member: 1
Comments to Author:

This manuscript addresses an interesting and important aspect of phenotypic plasticity. If the environment changes throughout development, and if organisms can alter their developmental trajectory continually in response to cues that are associated with these changes, what is the optimal developmental strategy? Unfortunately though, I think that R1 has identified a critical problem with the model. Undoubtedly the order in which the environmental states occur will affect the optimal developmental strategy but the model does not account for this. Instead, it assumes that all that matters is the number of times the environment is in each state during development but not the order in which they occur. I think this excludes a fundamentally important feature of the problem and so, unless we have both missed something, a different model would be required to analyze this question in a more satisfying way.

REPLY: We agree with the editor that R1's suggestion of using the sequential order of cues as the informational state is the better methodological choice. We have therefore redone the model,

reproduced all analysis in the main text and ESM with the corrected informational state, and adjusted the manuscript where necessary to account for the changes in methods and results. However, we would like to note that it is not accurate to state that our previous approach did not account for the order of cues. To compute the informational state (posterior estimates) we considered that different sequences have different likelihoods of occurring when the environment changes. R1 seems to think that we assign the same likelihood of occurrence for each underlying sequence of cues, as would be appropriate in a stable environment. We provide formulas and a detailed description of our approach in ESM 3.b (Hidden Markov Model and forward algorithm; both in previous and current submission). Due to space constraints we were not able to include these details in the main manuscript. However, we did refer to the relevant sections in the ESM where appropriate. After carefully considering R1's feedback, we acknowledge that computing posteriors for each sequential order of sampled cues is the better approach. Nonetheless, the approaches are mathematically related because both consider the order of cues when computing posteriors. Therefore, it is not surprising that we are able to replicate our main findings. That said, some effects are stronger than they initially were. For example, we had reported that critical periods, after which plasticity closes, are unlikely to be favoured in environments that fluctuate across ontogeny. This finding replicates. Also, levels of plasticity at the end of ontogeny are now higher than before, sometimes even resulting in sensitive periods at the end of ontogeny.

XX

REVIEWER COMMENTS

XX

Reviewer 1

This ms provides a simple model that attempts to understand developmental plasticity when changes during development are incremental. The effects of incremental change are worth exploring as nature works in that way. Of course the model has no costs of change, and so is rather limited. Nevertheless, one has to start somewhere, so I am not suggesting that costs should be introduced. From the current ms, I do not have a sense of how allowing changes to occur faster would change results. It would be good to have that intuition. The ms also varies the time horizon after maturation, which can then be used to explore how the likelihood of change after maturation effects results. This is an interesting issue. I am not sure that varying the time horizon is the best way to get at this question, but it is one way. Overall the general approach of the ms has the potential to make a worthwhile contribution to the literature. However, there appears to be one serious problem with the analysis (see directly below) and various other issues that require attention.

REPLY: We thank the reviewer for the positive assessment of our research question. While it is true that our model does not incorporate costs for making phenotypic adjustments or sampling cues, it does “assume that plasticity is incremental and irreversible, and that there is a cost to phenotype-environment mismatch in adulthood (lines 384-386)”. We elaborate on these costs in the Model section under ‘(d) Fitness during adulthood’.

Our model provides insights about how the rate of environmental change shapes levels of plasticity across ontogeny. As one of the main takeaways, our model suggests that plasticity levels at the end of ontogeny are typically higher when the environment changes faster. We have tried to better convey this insight in our resubmission. In the Results section we write “Lower autocorrelations typically result in higher levels of plasticity at the end of ontogeny (figure 2). The more frequent environmental

fluctuations are, the more cues can shift posterior estimates throughout all of ontogeny, increasing the scope for plasticity (figure 1) (lines 314-316)”. We have also incorporated this result in the discussion section: “With these assumptions, the level of plasticity at the end of ontogeny is highest when adulthood is short and the rate of environmental fluctuations is high (lines 386-387)”.

A problem with the state representation. In order to calculate the optimal strategy under dynamic programming the model takes as its informational state variable $D = (c_0; c_1)$, where c_0 and c_1 are the number of times each cue has been received so far. If I have understood the analysis in the paper correctly, the order of the cues is not important. Thus, say, receiving 5 C_0 followed by 5 C_1 is taken to be equivalent to receiving 5 C_1 followed by 5 C_0 . However, these two sequences lead to different posterior probabilities at time $T = 10$. This is because the environment is changing, implying that more weight should be placed on more recent cues. I may have misinterpreted the model, but I do not think so. For example under my interpretation there should be 11 possible D values at final time $T = 10$, and this appears to be the case in the figures. If instead, the order is taken into account then we would expect $2^{10} = 1024$ possible states at time $T = 10$. Ignoring the order will completely distort results. So if I am correct, I am afraid the whole model needs to be changed. To model the situation taking the order into account one could take the ordered sequence of previous cues as the current informational state. Alternatively one could just take the Bayes posterior as the state variable. This latter approach is certainly conceptually simpler. The drawback is that the posterior lies on a continuum, so that a fine grid with grid interpolation is necessary in order to do dynamic programming. Nevertheless, one should be able to dynamic program over many time steps relatively quickly.

REPLY: We thank the reviewer for identifying this methodological flaw in our previous submission. It is correct that the informational state variable in our previous model corresponded to $D = (c_0; c_1)$, where c_0 and c_1 are the number of times each cue has been sampled. However, it is not accurate to state that we did not take the underlying order of cues into account. When computing posterior estimates for a cue set $D = (c_0; c_1)$, we evaluated all possible orderings of cues that result in the total of c_0 and c_1 . Our approach considered that different sequences have different likelihoods of occurring when the environment changes. We had provided formulas and a detailed description of our approach in ESM 3.b (Hidden Markov Model and forward algorithm; both in the current and previous submission). Due to space constraints we were not able to include these details in the main manuscript. However, we did refer to the relevant sections in the ESM where appropriate. After carefully considering the feedback, we acknowledge that computing posteriors for each sequential order of sampled cues is the better approach. We have therefore redone the model, reproduced all analysis in the main text and ESM with the corrected informational state, and adjusted the manuscript where necessary to account for the changes in methods and results.

As a result of these changes, section ‘(e) Optimal developmental policies’ in the Model section now reads:

“To obtain optimal policies, we use posterior estimates across ontogeny to compute expected fitness across adulthood. We treat the states of the environment during ontogeny as ‘hidden’, unobserved states and sampled cues as ‘observed’ states of a Hidden Markov Model. We then apply the forward algorithm to compute the posterior probabilities, $P(E_0|D_t)$ and $P(E_1|D_t)$ for all possible orderings of sampled cues D_t [33]. $D_t = \{x_1, x_2, \dots, x_t\}$ denotes the sequence of cues until time period t , where x_1, x_2 , and so forth until x_t denote the cue (C_0 or C_1) sampled in each time period. We provide the formulas of the forward algorithm in ESM 3 (lines 169-175)”.

As indicated earlier, our previous and current approach are mathematically related because both consider the order of cues when computing posteriors. Therefore, it is not surprising that we are able to replicate our main findings. Some effects are even stronger than they initially were. For example, we had reported that critical periods, after which plasticity closes, are unlikely to be favoured in environments that fluctuate across ontogeny. This finding replicates. Moreover, levels of plasticity at the end of ontogeny are now higher than before.

As a result of these changes we now present sensitive periods at the end of ontogeny as a finding in the Results section.

“Sensitive periods often evolve towards the end of ontogeny. Frequent environmental fluctuations favour sensitive periods towards the end of ontogeny. In such conditions (autocorrelations of 0.2 and 0.5), organisms specialize according to the most recent cues prior to the onset of adulthood (figures 1-2). When environmental fluctuations are rare (autocorrelation of 0.8), plasticity sometimes peaks towards the end of ontogeny. When the adult lifespan is moderate (5 time periods) and cues are moderately reliable (0.75), a small proportion of the population specializes towards the less likely state in later time periods, resulting in sensitive periods towards the end of ontogeny. Plasticity may also peak at the end of ontogeny when the adult lifespan is short (1 time period) and cues are highly reliable (0.95), because organisms always choose to specialize according to cues in the final time period (figures 1-2). These are also the only conditions in our model that favour two peaks in plasticity: one smaller peak halfway through ontogeny and one larger peak in the final time period. In the second half of ontogeny, plasticity decreases because many organisms are locked into developmental trajectories on which they consistently specialize towards the same state. However, to reduce mismatch penalties during a short adulthood, organisms always specialize according to the final cue as a form of insurance (lines 343-356)”.

Other major points

- The motivation for the two states is that one might be safe and the other dangerous (lines 144-145). Furthermore, induced defences (as in *Daphnia*) provide the motivation for specialisation. Strictly speaking, the current model is not applicable in this case as the cost of having low induced defences is that the organism expects a shorter life. However, there is no mortality in the model and individuals all have exactly the same lifespan regardless of their strategy and the environment. I am not suggesting altering the model to include mortality. However, the authors cannot just use the induce defence example as motivation without a ‘health warning’.
- There is insufficient motivation of the payoff structure, even in the case of a linear payoffs. Consider the linear case. In this case the terminal reward at maturation is a linear function of $y_1 - y_0$ (equation 11 below). This fact is not brought out clearly. In this case one can try to give a motivation in terms of the induced defence example: y_1 is investment in defence, y_0 is removing existing investment. So the final $y_1 - y_0$ is the defence at maturity. Is this reasonable - maybe. The important point is that the ms does not say enough about the biological interpretation of the reward structure, even in this linear case. Note that in the linear case one can just use $z = y_1 - y_0$ as the physiological state variable in the dynamic programming. This reduces the dimension by 1.
- So what about the motivation for the non-linear case. Now the terminal reward is a function of $f(y_1) - f(y_0)$. How are we to interpret this? For the functions used in the non-linear case the terminal reward is no longer a function of $y_1 - y_0$. So what does this mean? It is like there are two traits that control the vulnerability to predation??

REPLY: We thank the reviewer for bringing these issues to our attention and for challenging our assumptions. We think it is helpful to address these three points raised by the reviewer together. To start with we have made more explicit that the traits we have modelled are not arranged on a continuum. In our model, phenotypic adjustments towards one phenotypic target do not reduce phenotypic adjustments towards the other target. We have now made this explicit in the Model section:

“For each environmental state, there is a corresponding optimal phenotype: P_0 for E_0 (e.g. specialized for foraging seeds) and P_1 for E_1 (e.g. specialized for foraging fruits). These two phenotypes are not arrayed along a single dimension, but along two, orthogonal dimensions. That is, increments toward one phenotypic target are not moves in the opposite direction of the other target, but in a different direction (lines 126-129)”.

We agree with the reviewer that our examples could be misleading. Using induced defences to predators as an example suggests that our phenotypic targets lie on a continuum and that our model considers mortality, neither of which is true. To illustrate our approach with a better example we now use being specialized for foraging fruits and being specialized for foraging seeds (see quote from the manuscript above). We now also make explicit that our model does not include mortality in the Model section.

“In this model, fitness is only a function of fertility. We consider the effects of viability selection in the Discussion section (lines 136-137)”.

Additionally, we have added a discussion of fitness as a function of mortality, both as a limitation of our current model and as a potential future direction.

“In our model, fitness is only a function of fertility. Other state-dependent models assume that fitness depends on fertility and mortality [16, 45]. Our model could be extended to include mortality. Mortality would be a function of phenotype-environment match during ontogeny, adulthood, or both, depending on how the trait influences mortality across these stages (lines 468-471)”.

It is true that we could mathematically express fitness as a function of $(y_0 - y_1)$ in the case of linear reward and linear penalties. Both our current formula and the shorter version using the difference $(y_0 - y_1)$ are equivalent and, therefore, yield the same numerical results. Using the difference would allow us to reduce the number of phenotypic dimensions to 1. We decided against making this adjustment for two reasons. First, we ultimately want to be able to compute optimal policies and changes in plasticity for non-linear reward and penalty mappings. While it may be possible to reduce the number of dimensions even in that case, the benefit of having a simple expression of the difference $(y_0 - y_1)$ would be lost. Second, reducing the number of dimensions and using the difference $(y_0 - y_1)$ in place of the individual components implies that the difference itself has a meaningful interpretation. If y_0 and y_1 would be arranged on a continuum, the difference is meaningful. However, if these phenotypic targets are independent the difference is less meaningful. A possible interpretation may be that this difference indicates how many increments more the organism has done towards one phenotypic target relative to the other. To us using $(y_0 - y_1)$ as the one dimension in our model then would make it more difficult to interpret some of our results, such as the distribution of mature phenotypes (ESM7, figure E7.38-E7.45). If we would reduce the number of phenotypic dimensions to one, we are also not sure how we would change the phenotypic decisions available to the organism. If our traits were arranged on a continuum and we imagine this continuum as a slider, phenotypic decisions in each time period could then be to wait (and forgo specialization), or to move the slider one step to the right, or the left. However, with independent, orthogonal traits, we are not sure how to conceptualize the phenotypic decision (aside from waiting) available to the organism. Considering all of the above we still think

that retaining both phenotypic dimensions in the model is better suited to explore our research interests. We are, however, grateful that the reviewer showed us the potential for reducing the complexity of our model and will consider it for future work.

We hope that our updated examples and more explicit discussion of assumptions resolve the issues raised by the reviewer.

- The autocorrelation was found by simulation, whereas it is easy to calculate. I derive the formula in the technical details below (equation 7). This formula should be in various elementary texts on time series so the authors should just quote it and do not need to derive it. It might be worth quoting this simple formula in the main text in order to give the reader a better intuition.

REPLY: We thank the reviewer for questioning our use of simulations to compute autocorrelations in the asymmetric case. It helped us realize that the formula we applied in the symmetric case is a special case of the formula suggested by the reviewer. We had reported the formula for the symmetric case in ESM 4 of our original submission. Based on previous correspondence with a leading expert in applying stochastic dynamic programming to evolution in fluctuating environments, we reached the conclusion that this formula would not generalize to the asymmetric case. The reviewer pointed out that this is in fact possible for which we are grateful. We can confirm that the formula yields the exact same outcome as our simulation did for the asymmetric case. We now use the formula in our code and also adjusted ESM 4.

Minor points

- Lines 124-128. The transition probabilities are over unit time. However, this is not said and the time structure has yet to be introduced.

REPLY: Following the reviewer's suggestion, we now introduce the time structure in the Model section when we first discuss transition probabilities.

“From one time period to the next, the state of each patch switches stochastically between E_0 and E_1 with transition probabilities, $P(E_{0,t}|E_{1,t-1})$ and $P(E_{1,t}|E_{0,t-1})$ where t denotes the current time period. For example, a patch might start out rich in one food type and switch to a different food type (e.g. seeds or fruits). We use a Markov process to fully describe the transitions between states. As the per time period transition probabilities are fixed, we abbreviate $P(E_{0,t}|E_{1,t-1})$ and $P(E_{1,t}|E_{0,t-1})$ with $P(E_0|E_1)$ and $P(E_1|E_0)$ (lines 109-114)”.

- Might it be worth plotting the terminal reward?

REPLY: We already plot the fitness of the optimal policy for all reward and penalty mappings including the linear case in ESM7 (figures E7.46 - E7.54).

- It appears that in looking at the sensitivity to cues, the cues in the mirror patch are opposite those in the original patch from a given time onwards. If that is true I personally find the measure rather bizarre. Surely it would be more informative just to flip one cue, keeping cues before and after the same?

REPLY: We chose this measure in line with previous models as a way to resemble study paradigms used in empirical studies. We added the following text to better motivate our choice:

“We simulate experimental designs resembling empirical adoption studies. These studies compare mature organisms, often twins or siblings, separated at a particular point during ontogeny for a specific duration. Researchers investigate how the age at which organisms are separated (and possibly later reunited), and the conditions during separation, determine variation in mature phenotypes. We have previously shown that different manipulations of experiences during separation – for instance, receiving reciprocal opposite cues or cues from a different patch, and temporary or permanent separations – yield similar qualitative patterns [32]. These patterns are most pronounced for reciprocal opposite cues and permanent separation, as experience is maximally divergent for a longer time. Therefore, we analyze only this manipulation here (lines 206-213)”.

- The figure captions are not very informative as to underlying parameter values.

REPLY: All figures explicitly state the values of all parameters involved, such as the autocorrelation, cue reliability, and adult lifespan. In addition, each caption states the reward and penalty mapping and the number of simulations (if applicable). We are not sure what other parameter values the reviewer is referring to.

- Lines 283-284. It says that ‘Pies highlight cases in which organisms with the same estimates make different phenotypic decisions’. I am unclear what this means.

REPLY: Organisms with the same posterior estimates can make different phenotypic decisions if they reach those posteriors with different phenotypes. To make this more explicit, we rewrote the sentence. It now reads: “Pies highlight cases in which organisms with the same posterior estimates (but different phenotypic states) make different phenotypic decisions (lines 276-278)”.

- The model has just two environmental states, so that the Bayes posterior is characterised by a single variable so that the posterior variance is a function of the posterior mean. In contrast, if there were a continuum of environmental state, the mean does not determine the variance. One can then have the posterior variance decreasing over time even though the mean stays roughly the same. Prediction will then be altered. It might be worth drawing this to readers’ attention, and perhaps mentioning different biological scenarios that illustrate this.

REPLY: We thank the reviewer for this suggestion. We have added the following text to the Discussion section under ‘(d) Limitations and future directions’:

“Our model assumes only two environmental states. Another possibility would be to assume a larger number of discrete states or a continuum of states. This would make it possible to independently manipulate the mean and variance of the prior distribution over states, as well as the reliability of cues. Doing so might alter the findings from our model. However, previous models of stable environments which incorporate a continuum of environmental states [13, 15], as well as those assuming two discrete states [11, 12], have found similar qualitative patterns. Future modelling could explore whether our results replicate when increasing the number environmental states (lines 461-467)”

Technical details which may be useful to the authors

Warning: I have used a different notation to the ms in what follows. That is because I personally find the standard mathematical notation I use clearer. I am not suggesting the authors adopt this notation.

Let $X_t = 0$ if the environmental state at time t is E_0 , with $X_t = 1$ if the environmental state at time t is E_1 . I use the notation that $P(X_{t+1} = 1|X_t = 0) = \lambda_1$ and $P(X_{t+1} = 0|X_t = 1) = \lambda_0$.

Posterior distribution changes with no cues.

Let $p(t) = P(X_t = 1)$. Then $p(t + 1) = (1 - p(t))\lambda_1 + p(t)(1 - \lambda_0)$: (1) To find the stationarity probability we set $p(t + 1) = p(t) = p^*$ in equation 1 to give $p^* = \lambda_1/\lambda_0 + \lambda_1$: (2)

From equation 1 we can then get

$$p(t + 1) - p^* = \rho [p(t) - p^*]; (3)$$

where

$$\rho = 1 - (\lambda_0 + \lambda_1) (4) \text{ is the autocorrelation (see below).}$$

$$\text{Applying this formula recursively gives } p(t) = p^* + \rho^t [p(0) - p^*]; (5)$$

Autocorrelation

Note that at stationarity we have $E(X_t) = E(X_{t+1}) = p^*$. We also have $E(X_t X_{t+1}) = P(X_t = 1; X_{t+1} = 1) = P(X_t = 1)P(X_{t+1} = 1|X_t = 1) = p^*(1 - \lambda_0)$. From these equation we get

$$\text{Cov}(X_t; X_{t+1}) = E(X_t X_{t+1}) - E(X_t)E(X_{t+1}) = p^*(1 - \lambda_0 - p^*): (6)$$

It is also easy to see that $\text{Var}(X_t) = \text{Var}(X_{t+1}) = p^*(1 - p^*)$. Thus the correlation between X_t and X_{t+1} is $\rho = \text{Cov}(X_t; X_{t+1}) / \sqrt{\text{Var}(X_t)\text{Var}(X_{t+1})} = 1 - (\lambda_0 + \lambda_1)$: (7)

The terminal reward

To simplify notation let $p(t)$ denote the probability that the environment is E_1 after t time steps in the adult phase, conditional on the information available at the end of the juvenile phase: in the authors' notation $p(\text{tadult}) = P(E_1; \text{tadult} | \text{tjD})$. Thus $p(0)$ is the posterior at the end of the juvenile phase and $p(t)$ is given by equation 5.

We can write line 339-340 of ESM as

$$\varphi(y_0; y_1; t) = (1 - p(t))f(y_0) + p(t)f(y_1) (8)$$

$$(y_0; y_1; t) = - [(1 - p(t))f(y_1) + p(t)f(y_0)]: (9)$$

$$\text{Thus } \varphi(y_0; y_1; t) + (y_0; y_1; t) = [f(y_1) - f(y_0)] (2p(t) - 1): (10)$$

It then follows that fitness can be expressed as $\pi(y_0; y_1) = \pi_0 + [f(y_1) - f(y_0)] L$; (11) where L depends on $p(t) : t = 0; 1; \dots; \text{Tadult}$. Equation equation 5 can then be used to find an analytic expression for L (by just summing a geometric progression). This expression might be a bit messy in general, but is especially simple in the symmetric case.

REPLY: We thank the reviewer for these helpful details which we have considered when revising our manuscript. Thanks to these technical details we were able to replace our simulation approach for computing autocorrelations with a more general formula. We also carefully studied the fitness derivations and are considering whether or not to reduce the number of dimensions in our model for future work. Thank you.

Reviewer 2

The manuscript describes a model that is used to investigate which conditions favour critical or sensitive periods during development for plastic adjustments to the phenotype. Overall, I think the model results are interesting and the manuscript is well written.

REPLY: We thank the reviewer for the positive assessment of our work.

I therefore have only a few comments and suggestions for the authors:

1. 45-47: It is a bit difficult to understand what critical periods in this sentence alone. I also find this sentence to not be quite necessary and would suggest you cut this. If not, please rephrase.

REPLY: We have rewritten our abstract and now better explain the distinction between sensitive and critical periods. The relevant text now reads:

“We conclude that critical periods, after which plasticity is zero, are unlikely to be favoured in environments that fluctuate across ontogeny (lines 45-46)”.

1. 71/75: Again, the definition of “critical periods” is not quite clear. In order to accommodate a wider audience, I think it is necessary to explain this a bit further before this term is used as here.

REPLY: We agree with the reviewer’s suggestion. Similarly to our previous adjustment we have rewritten the text to provide a clearer definition of critical periods. The text now reads:

“Such a pattern would differ from that observed in models of stable environments, which often favour critical periods, where plasticity drops to zero [17, 18] (lines 67-68)”

1.113 and after: You state that phenotypic change is irreversible, but the way I understand your model an individual that has specialized once towards each environmental state is back to start. If this is correct then the model would be equivalent to one where phenotypic change is reversible. This is mentioned several times and is an important point in the discussion as well, so some further clarification is strongly encouraged.

REPLY: The reviewer’s interpretation of phenotypic development in our model was not correct. Organisms incrementally and irreversibly develop phenotypic adjustments. Once an adjustment has been made, it cannot be undone. For simplicity, let us ignore the fact that organisms can wait and only consider the two phenotypic dimensions, Y_0 and Y_1 . At the start an organism’s phenotype is $(y_0 = 0, \text{ and } y_1 = 0)$, where y_0 and y_1 refer to the number of increments made towards each phenotypic target. Thus, an organism which has developed once towards each state is not back to the start. Her phenotype would correspond to $(y_0 = 1, y_1 = 1)$. The organism’s posterior estimate will also have changed from its prior estimate after having made these two phenotypic adjustments. We have rewritten parts of the Introduction, Model, and Discussion section to better explain in what way our model considers development to be irreversible:

“Once phenotypic increments have developed, they cannot be undone. (lines 97-98)”.

“After each cue, organisms have three options: specialize one increment towards P_0 , specialize one increment towards P_1 , or wait and forgo specialization. Once an increment has developed, it cannot be undone, yet organisms may always switch developmental trajectories (lines 130-133)”.

“The ability to reverse development may reduce phenotype-environment mismatch and thus make plasticity across all of ontogeny more viable. In our model, organisms cannot reverse phenotypic increments. Developmental trajectories, however, are reversible, such that organisms may specialize towards the opposite phenotypic target at any point during ontogeny (lines 389-393)”.

l. 228/239-240: The point of, and therefore the meaning of “ t ” is unclear on line 228, maybe move the info on l 139-240 up?

REPLY: Following the reviewer’s suggestion we have restructured the order of information presented in the Model section under ‘(b) Quantifying plasticity’. The relevant text not starts by introducing what we mean by t :

“We use the optimal policy to simulate developmental trajectories. The level of plasticity corresponds to the extent to which phenotypic development depends on cues during ontogeny. We compute plasticity for each $t \in \{1, T_{ont}\}$. We start by simulating pairs of clones, following the optimal policy. Organisms start in either environment, E_0 or E_1 . We simulate all possible sequences of cues, resulting in one pair of clones per sequence. Each pair of clones receives a weight according to the likelihood of its particular cue sequence.

Clones develop together until time period t , experiencing the same sequence of cues and making the same phenotypic decisions, resulting in identical phenotypes (lines 214-220)”.

236-240: Your measure of stage-specific plasticity is very important for the understanding of the results, and seems appropriate to me, but it took me a long time to understand what this measure is and how it is calculated, so I think it is necessary with a better explanation of this. A figure might be helpful here.

REPLY: We added a schematic overview of our simulated adoption paradigm in ESM 6 (figure E6.1) and refer to it in the Model section.

Figure1:

Y-axis label, left side (“estimate”) is not very explanatory, could you relabel?

REPLY: We changed the label for the figure in the main text and all figures in the ESM to $P(E_1|D)$, i.e. the organism’s posterior estimate.

L 279-280. I presume this is the estimate of the probability of being in E_1 ?

REPLY: It is the posterior probability of being in E_1 .

l. 281/282: please specify what the prior and posterior refer to.

REPLY: We explicitly added that prior and posterior in the figure legend are referring to E_1 now.

“The vertical axis displays posterior estimates of being in E_1 and the horizontal axis displays time during ontogeny. At the onset of ontogeny, all organisms start with a prior estimate of being in E_1 according to the stationary distribution (large grey circle) (lines 272-274)”.

l. 284: I can’t see black appear in this graph?

REPLY: Black indicates that organisms chose to wait, which they never do for linear rewards and penalties presented in the main text. We altered the legend to make this more explicit:

“Pies highlight cases in which organisms with the same posterior estimates (but different phenotypic states) make different phenotypic decisions. Black corresponds to waiting (not visible here because organisms never choose to wait), blue to specializing towards P_1 , red to specializing towards P_0 (lines 276-279)”

Figure 2:

This figure is possibly more confusing than helpful. You seem to need more dimensions than available and therefore have adult lifespan nested within transition probability and autocorrelation nested within cue reliability. While this might be necessary to show all results in one figure, it needs to be much clearer that the two major axes are not at all continuous.

Since the axes are not continuous, the areas with black frames does not make complete sense to me. There is not reason why asymmetric E0 should be left of asymmetric E1, and the picture would look quite different if these switched places. It would also be possible to nest e.g. transition probabilities within adult lifespan, which would also completely change the picture. A figure that makes the nesting more obvious might be better, but it is not quite clear to me what the goal of this figure is.

REPLY: We agree with the reviewer’s assessment. We have removed the figure and restructured our Results section to be easier to follow, even without the additional visual guidance that the figure was supposed to provide.

Figure 3:

It is again a bit difficult to understand the measure on the y-axis and the label is not quite self-explanatory. Right now it looks like all phenotypes are always very similar at the end of ontogeny.

REPLY: The label indicates that the y-axis represents the phenotypic distance between pairs of simulated clones. We also explain this in detail in the figure legend (“We compute phenotypic distance (vertical axis) as the average, weighted Euclidean distance of all pairs of clones at the end of ontogeny and plot it against the time of separation. Phenotypic distance is normalized by dividing it by the maximally attainable Euclidean distance (lines 369-371)”). We hope that all of the adjustments that we made to the Model section, as well as the figure that we added to the supplements (E6.1) have increased clarity such that the interpretation of the figure and the figure legend has become much easier.

Appendix C

Comments on the revised version of Walasek et al.

I was Reviewer 1 on the previous submission to PRSB. In my previous review I raised three major points.

1. The state representation. In the previous version individuals were constrained to base their decision on the number of times each cue had been received so far. This meant that an individual had to take the same action if 4 C_0 cues were followed by 4 C_1 as when 4 C_1 cues were followed by 4 C_0 . This was not realistic. In the current version individuals are no longer so constrained, and can base their decision on the full sequence of cues received so far. This seems a much better way to model plasticity. The analysis for this case appears to be competently done (but see point 4 below).

2. The interpretation of costs and benefits. In the previous version I questioned the the motivation for the cost/benefit structure. In particular, I noted that the terminal reward was proportional to $f(y_1) - f(y_0)$ and asked the authors to provide suitable motivation for this assumption.

In the current version, certain modelling aspects are clearly expressed. For example, it is clear that we are dealing with two different traits, not a single trait that can increase or decrease. The authors use the word ‘orthogonal’ here, although this is a rather vague phrase (as oppose to the mathematical definition of orthogonality, which is precisely defined). It is also clear that each trait increases by increments, and that increases cannot later be undone. However, I believe that we are not given a strong enough connection between the traits and cost structure, and the underlying biology.

To illustrate this, I note that the modelling section motivates assumptions using the example of an environment that is either seed rich or fruit rich. In this environment one hypothetical trait prepares the organism for a seed rich environment, the other prepares it for a fruit rich environment. The author do not, however, tell us

what these traits might be biologically. If the traits involve the development of motor skills (a trait mentioned in the Introduction), then an organism that developed equal motor skills for both activities ($y_0 = y_1 > 0$ at maturity) would do equally well as an individual with zero skills in both ($y_0 = y_1 = 0$ at maturity) in both environments. This occurs since the terminal reward is proportional to $f(y_1) - f(y_0)$. That does not seem sensible. So perhaps the traits are to do with beak development in a bird? If so, there would have to be two different aspects of beak morphology that are ‘orthogonal’ and where one aspect is costly in the wrong environment. It would be good to have biological motivation that actually tied in to the model used.

These example, illustrate the restricted and specific nature of the model considered. A casual reader might well think the model is rather general since it considers both ‘linear’ and ‘non-linear’ cost. This is an illusion. A general model would allow the terminal reward to be an arbitrary function of (y_0, y_1) . A less general model would take the reward to be proportional to $f_1(y_1) - f_0(y_0)$, where f_0 and f_1 are distinct functions. Finally, we come to the model used here that restricts these two function to be equal. The authors justify the latter restriction on the ground that in a different model the restriction did not significantly change results. Maybe so, but maybe not.

I am not saying that I think that the authors need to make a general model. All models are limited, but their limitations need to be clear. Overall, readers need to better understand how the model relates to real biology. In that way a reader will be in a better position to assess what the model can tell us and what are its limitations.

3. Calculation of the autocorrelation. I am pleased that the formula I supplied was of use to you.

I have also noticed the following error in the current version

4. The dynamic programming equation. The dynamic programming equations

are given by ESM 3, lines 413-417. As stated, these equations give $F(D_t, \dots, t, T)$ in terms of $F(D_t, \dots, t + 1, T)$, whereas I would have expected the the latter expression to involve D_{t+1} . In other words, the equations as stated do not allow D_t to change with time. This is clearly wrong. I suspect that this is just a lack of careful presentation, and that the authors have used the correct DP equation, especially as they previously derive the relationship between D_{t-1} and D_t . I hope that this is so. However, I cannot check that until they actually write down the real equation that they used.

Minor points

The mathematical notation in the ESM 3 is very sloppy. It may only be the ESM, but people who really want to understand the results need to look here to find details, so this matters. In particular:

- Lines 298-299. What is $D_{1:t}$? Presumably it is D_t . What is $D_{i,1:t}$? Again it is presumably D_t . I also note that here you sum over st . Some readers may mistakenly think that $st = s \times t$. Why not just sum over i here.
- Both functions ψ and ϕ have two distinct definitions, in one they are functions of two arguments, then later they are a different function of one argument. Different function letters are needed for different functions.
- You state that t runs from 0 to T and that t_{adult} runs from 0 to T_{adult} . But then in your notation $E_{0,t}$ is not the same as $E_{0,t_{adult}}$ when, say $t = t_{adult} = 3$. In other words $E_{0,3}$ is not the same as $E_{0,3}$!! Of course many biologist may understand the sloppy notation, but that is not an excuse for using it. One way to avoid these problems is to have time run from time 0 (birth) to the end of the reproductive phase at time T_{end} . Maturity then occurs at time T , where $T + T_{adult} = T_{end}$.

REVISION

Sensitive periods, but not critical periods, evolve in a fluctuating environment: A model of incremental development

Dear Dr. Dall,

We would like you to consider our revision of “Sensitive periods, but not critical periods, evolve in a fluctuating environment: A model of incremental development” for publication in *Proceedings B* (RSPB-2021-2623). We are grateful for the opportunity to revise and resubmit our manuscript. We have addressed the reviewer’s concerns and feel the manuscript is stronger as a result of the revisions. In this letter, we list each of the comments from the editor or reviewer and address them in turn.

Sincerely,

Nicole Walasek¹, Willem E. Frankenhuis^{1,2,3} and Karthik Panchanathan⁴

¹ Behavioral Science Institute, Radboud University, the Netherlands

² Department of Psychology, Utrecht University, the Netherlands

³ Max Planck Institute for the Study of Crime, Security and Law, Germany

⁴ Department of Anthropology, University of Missouri, U.S.A.

This research was supported by grants from the Dutch Research Council (V1.Vidi.195.130) to WEF, the James S. McDonnell Foundation (<https://doi.org/10.37717/220020502>) to WEF, and the Jacobs Foundation (2017 1261 02) to WEF.

Correspondence concerning this article should be addressed to Nicole Walasek, Thomas van Aquinostraat 4, 6525 GD, Nijmegen, The Netherlands. Phone: +31629386516. E-mail: nicole.walasek@ru.nl.

Responses to editor and reviewer comments

The original comments are **black**. Our replies are **blue**. Text inserted from the manuscript is **green**.

XX

EDITORIAL LETTER

XX

Associate Editor Board Member

Comments to Author:

The authors have done a nice job of fixing the main problem with the previous model but there are a couple of important issues remaining that I think they need to address. Perhaps the most important is being clear about potential real-world biological scenarios where the model assumptions would apply. There are also some (hopefully minor) outstanding issues regarding the mathematical presentation.

REPLY: We agree with the editor and R1 that it is important to be clear and specific about which scenarios our model covers and which scenarios it does not. Based on R1's feedback, we replicated our analyses for different penalty weights (relative to rewards) and present these additional results in the ESM. As R1 pointed out, our original submission justified one set of penalty weights by referencing a previous model showing that different weights resulted in the same qualitative results. After running the current model across a range of weights, we find the same qualitative effects for both lower and higher penalty weights. Having extended our analyses with these additional penalty weights increases the generality of the current model. We added a brief description of the results for penalty weights 0.5 and 2 in the Discussion section.

“In our model, fitness is proportional to the difference between correct and incorrect specializations. We instantiate this through specific reward-penalty mappings and penalty weights. In the main text, we set the penalty weight to 1, implying that rewards and penalties contribute equally to fitness. In the ESM, we show that penalty weights of 0.5 and 2 yield the same qualitative patterns, but there is one difference: when the penalty weight is 2, organisms sometimes wait to reduce phenotype-environment mismatch. Though we have explored a wide parameter range, future work could investigate a more general model where fitness is an arbitrary function of phenotype. (lines 481-487)”

In addition, we added an example to the Methods section that makes our assumptions about costs and benefits more explicit and visible. This addresses R1's concern that even non-modelers should be able to understand the specific assumptions and limitations of our model.

“The fitness in each period of adulthood is calculated by summing the marginal rewards for correct specializations, marginal penalties for incorrect ones, and baseline fitness. We explore three mappings between phenotypes and marginal fitness rewards and penalties (linear, increasing, and diminishing) and three penalty weights (0.5, 1, and 2) [11, 12, 32]. The specific combination of mappings and penalty weight determines how organisms accrue fitness. Returning to our example of seed and fruit specialization, imagine the following two organisms: organism A has developed equal numbers of specializations for both states, while organism B has waited throughout ontogeny, developing zero specializations for either state. If we assume linear reward and penalty functions and a penalty weight of 1, then both organisms accrue zero fitness. If, instead, we assume a higher penalty weight or a

diminishing penalty function, then, all else equal, A would attain lower fitness than B. With a lower penalty weight or an increasing penalty function, B does better than A. In the paper we set the penalty weight to 1 and the reward and penalty mappings to linear. We present the other combinations in the ESM (ESM 7-8) and address them in the Discussion section. (lines 163-175)”

Lastly, we apologize for the inconsistent mathematical notation in the ESM. We have corrected this. We can assure you that R1’s interpretation of what the dynamic programming equations should look like is correct. Missing the ‘+1’ was indeed a typo. We fixed all points raised by R1 regarding our notation. We paste the revised notations below in response to R1’s comments.

XX

REVIEWER COMMENTS

XX

Reviewer 1

I was Reviewer 1 on the previous submission to PRSB. In my previous review I raised three major points.

REPLY: We thank the reviewer for providing another round of detailed comments. We appreciate these comments very much and describe below how we have incorporated them in our revision.

1. The state representation. In the previous version individuals were constrained to base their decision on the number of times each cue had been received so far. This meant that an individual had to take the same action if 4 C0 cues were followed by 4 C1 as when 4 C1 cues were followed by 4 C0. This was not realistic. In the current version individuals are no longer so constrained, and can base their decision on the full sequence of cues received so far. This seems a much better way to model plasticity. The analysis for this case appears to be competently done (but see point 4 below).

REPLY: Thank you.

2. The interpretation of costs and benefits. In the previous version I questioned the motivation for the cost/benefit structure. In particular, I noted that the terminal reward was proportional to $f(y1) - f(y0)$ and asked the authors to provide suitable motivation for this assumption. In the current version, certain modelling aspects are clearly expressed. For example, it is clear that we are dealing with two different traits, not a single trait that can increase or decrease. The authors use the word ‘orthogonal’ here, although this is a rather vague phrase (as oppose to the mathematical definition of orthogonality, which is precisely defined). It is also clear that each trait increases by increments, and that increases cannot later be undone.

REPLY: Thank you. To further clarify, we have (a) adopted the language of the reviewer (“two different traits; not a single trait that can increase or decrease”), (b) changed the word ‘orthogonal’ to ‘independent’, and (c) provided a description. This text now reads:

“These phenotypes represent two different traits, rather than a single trait that increases or decreases. In other words, phenotypes are not arrayed along a single dimension, but along two independent dimensions. Changes in one trait are independent of changes in the other trait. (lines 128-131)”

However, I believe that we are not given a strong enough connection between the traits and cost structure, and the underlying biology. To illustrate this, I note that the modelling section motivates assumptions using the example of an environment that is either seed rich or fruit rich. In this environment one hypothetical trait prepares the organism for a seed rich environment, the other prepares it for a fruit rich environment. The authors do not, however, tell us what these traits might be biologically. If the traits involve the development of motor skills (a trait mentioned in the Introduction), then an organism that developed equal motor skills for both activities ($y_0 = y_1 > 0$ at maturity) would do equally well as an individual with zero skills in both ($y_0 = y_1 = 0$ at maturity) in both environments. This occurs since the terminal reward is proportional to $f(y_1) - f(y_0)$. That does not seem sensible. So perhaps the traits are to do with beak development in a bird? If so, there would have to be two different aspects of beak morphology that are ‘orthogonal’ and where one aspect is costly in the wrong environment. It would be good to have biological motivation that actually tied in to the model used. These examples illustrate the restricted and specific nature of the model considered. A casual reader might well think the model is rather general since it considers both ‘linear’ and ‘non-linear’ cost. This is an illusion. A general model would allow the terminal reward to be an arbitrary function of $(y_0; y_1)$. A less general model would take the reward to be proportional to $f_1(y_1) - f_0(y_0)$, where f_0 and f_1 are distinct functions. Finally, we come to the model used here that restricts these two functions to be equal. The authors justify the latter restriction on the ground that in a different model the restriction did not significantly change results. Maybe so, but maybe not. I am not saying that I think that the authors need to make a general model. All models are limited, but their limitations need to be clear. Overall, readers need to better understand how the model relates to real biology. In that way a reader will be in a better position to assess what the model can tell us and what are its limitations.

REPLY: We agree with the reviewer that it is important to be explicit about the specific assumptions and limitations of our model. In particular, we can make our assumptions about costs and benefits of phenotypic specialization more explicit so that a ‘casual’ reader (i.e. a non-modeller) can more easily apply our model when thinking about specific, real-world biological examples. To address the reviewer’s concerns and improve the paper, we made two revisions.

First, we have added examples in the Methods section to illustrate how fitness depends on the relative weights of costs and benefits. As the reviewer notes, for the specific case of linear rewards and linear penalties, with equal reward and penalty weights, “an organism that developed equal motor skills for both activities ($y_0 = y_1 > 0$ at maturity) would do equally well as an individual with zero skills in both ($y_0 = y_1 = 0$ at maturity) in both environments”. We now make this explicit as follows:

“The fitness in each period of adulthood is calculated by summing the marginal rewards for correct specializations, marginal penalties for incorrect ones, and baseline fitness. We explore three mappings between phenotypes and marginal fitness rewards and penalties (linear, increasing, and diminishing) and three penalty weights (0.5, 1, and 2) [11, 12, 32]. The specific combination of mappings and penalty weight determines how organisms accrue fitness. Returning to our example of seed and fruit specialization, imagine the following two organisms: organism A has developed equal numbers of specializations for both states, while organism B has waited throughout ontogeny, developing zero specializations for either state. If we assume linear reward and penalty functions and a penalty weight of 1, then both organisms accrue zero fitness. If, instead, we assume a higher penalty weight or a

diminishing penalty function, then, all else equal, A would attain lower fitness than B. With a lower penalty weight or an increasing penalty function, B does better than A. In the paper we set the penalty weight to 1 and the reward and penalty mappings to linear. We present the other combinations in the ESM (ESM 7-8) and address them in the Discussion section. (lines 163-175)”

Second, we ran additional analysis for different penalty weights to increase the generality of our model. It is true that we did not explore different penalty weights in the previous submission. We only referenced a related model that found no qualitative differences in results for different weights. Based on the reviewer’s concerns, we ran the model with different penalty weights. Specifically, we ran analyses with penalty weights of 0.5 and 2. Consistent with the previous model, we did not find qualitative differences. We show the resulting plots in the ESM (figures E8.1-E8.6) and briefly comment on our findings in the Discussion section:

“In our model, fitness is proportional to the difference between correct and incorrect specializations. We instantiate this through specific reward-penalty mappings and penalty weights. In the main text, we set the penalty weight to 1, implying that rewards and penalties contribute equally to fitness. In the ESM, we show that penalty weights of 0.5 and 2 yield the same qualitative patterns, but there is one difference: when the penalty weight is 2, organisms sometimes wait to reduce phenotype-environment mismatch. Though we have explored a wide parameter range, future work could investigate a more general model where fitness is an arbitrary function of phenotype. (lines 481-487)”

This addition makes our model more general as we now also explore cases in which an organism that has developed equal motor skills for seed- and fruit-rich environments ($y_0 = y_1 > 0$ at maturity) does better (or worse) than an individual with zero skills ($y_0 = y_1 = 0$ at maturity) in both environments.

Lastly, we do not agree with the reviewer that our model is limited to the case where the reward and penalty mappings have the same functional forms (“A less general model would take the reward to be proportional to $f_1(y_1) - f_0(y_0)$, where f_0 and f_1 are distinct functions. Finally, we come to the model used here that restricts these two function to be equal.”). In all our previous submissions and the current one, we refer the reader to the ESM for different combinations of reward and penalty mappings. We explore all possible combinations of linear, increasing, and diminishing rewards and penalties.

“We explore three mappings between phenotypes and marginal fitness rewards and penalties (linear, increasing, and diminishing) and three penalty weights (0.5, 1, and 2) [11, 12, 32]. (lines 165-167)”

Our model does not offer a general, analytical solution that applies to all scenarios. However, we do present a large range of scenarios in the ESM and even added two additional scenarios as part of this revision. We made the choice to present the most ‘basic’ case of linear rewards and linear penalties with equal reward and penalty weights in the manuscript as it is consistent with previous models and made sense to us. However, some readers may be interested in the findings for different parameter settings. Presenting and discussing all of those in the main text is not feasible given space restrictions. The same would be true for a general, analytical solution. By presenting additional parameter combinations in the ESM, our revision provides insights into the evolution and development of sensitive periods for a broader range of traits and not only those specific cases highlighted by the reviewer.

3. Calculation of the autocorrelation. I am pleased that the formula I supplied was of use to you.

REPLY: We thank the reviewer for this excellent suggestion.

I have also noticed the following error in the current version:

4. The dynamic programming equation. The dynamic programming equations are given by ESM 3, lines 413-417. As stated, these equations give $F(Dt; ; ; ; t; T)$ in terms of $F(Dt; ; ; ; t + 1; T)$, whereas I would have expected the the latter expression to involve $Dt+1$. In other words, the equations as stated do not allow Dt to change with time. This is clearly wrong. I suspect that this is just a lack of careful presentation, and that the authors have used the correct DP equation, especially as they previously derive the relationship between $Dt-1$ and Dt . I hope that this is so. However, I cannot check that until they actually write down the real equation that they used.

REPLY: We thank the reviewer for spotting this. We can assure you that the reviewer's interpretation of what the dynamic programming equations should look like is correct. Missing the '+1' was indeed a typo. We paste the revised notation below.

" $F(D_t, y_0, y_1, y_w, t, T_{ont})$ denotes the maximum expected fitness that can be attained across adulthood as a result of decisions made between t and T_{ont} , when the organism's current state after the last cue sampled is $(D_{T_{ont}}, y_0, y_1, y_w, T_{ont})$ and the organisms chooses option a , so that:

$$F(D_t, y_0, y_1, y_w, t, T_{ont}) = \max_{a \in \{0,1,w\}} F_a, \text{ where}$$

$$F_0 = F(D_{t+1}, y_0 + 1, y_1, y_w, t + 1, T_{ont}),$$

$$F_1 = F(D_{t+1}, y_0, y_1 + 1, y_w, t + 1, T_{ont}),$$

$$F_w = F(D_{t+1}, y_0, y_1, y_w + 1, t + 1, T_{ont}).$$

We apply backwards induction to solve the dynamic programming equation $F(D_t, y_0, y_1, y_w, t, T_{ont})$ for all t . We start with $t = T_{ont}$:

$$F(D_{T_{ont}}, y_0, y_1, y_w, T_{ont}, T_{ont}) = \pi_{Total}(Y_{mat}).$$

After calculating expected fitness at the end of ontogeny we continue by decrementing t . For each $t < T_{ont}$ we compute the a , which maximizes $F(D_{t+1}, y_0, y_1, y_w, t + 1, T_{ont})$ in time period t . (ESM 3, lines 410-423)"

Minor points

The mathematical notation in the ESM 3 is very sloppy. It may only be the ESM, but people who really want to understand the results need to look here to find details, so this matters. In particular:

- Lines 298-299. What is $D1:t$? Presumably it is Dt . What is $Di;1:t$? Again it is presumably Dt . I also note that here you sum over st . Some readers may mistakenly think that $st = s \times t$. Why not just sum over I here.

REPLY: We thank the reviewer for spotting this. We wanted to convey that the recursion runs from 1 to t but we already say as much in the text. Indeed, just Dt is the most appropriate representation. We agree that st may be confusing and replaced it with i . We paste the revised notation below.

“For each possible sequence $D_t = \{x_1, x_2, \dots, x_t\}$ of cues C_0 and C_1 , we apply the forward algorithm as defined by the following recursions running from 1 until the current time period t during ontogeny:

$$P(E_{0,t}, D_t) = P(x_t|E_{0,t}) \cdot \sum_{i \in \{0,1\}} P(E_{i,t-1}, D_{t-1}) \cdot P(E_{0,t}|E_{i,t-1})$$

$$P(E_{1,t}, D_t) = P(x_t|E_{1,t}) \cdot \sum_{i \in \{0,1\}} P(E_{i,t-1}, D_{t-1}) \cdot P(E_{1,t}|E_{i,t-1})$$

$P(x_t|E_{0,t})$ and $P(x_t|E_{1,t})$ are specified by the cue reliabilities $P(C_0|E_0)$ and $P(C_1|E_1)$, depending on whether the t^{th} cue in the sequence corresponded to C_0 or C_1 . $P(E_{0,t}|E_{i,t-1})$ and $P(E_{1,t}|E_{i,t-1})$ are specified by the transition probabilities between states, where $E_{i,t-1}$ can correspond to either E_0 or E_1 . (ESM 3, lines 293-303)”

• Both functions and φ have two distinct definitions, in one they are functions of two arguments, then later they are a different function of one argument. Different function letters are needed for different functions.

REPLY: We thank the reviewer for spotting this. In our revision, we use a different function for total fitness across adulthood and fitness at any time period in adulthood, as described below.

“

Functions and constants	Explanation
$\phi(Y_{mat}, t)$	Expected, fitness reward at time period t during adulthood ($t > T_{ont}$)
$\psi(Y_{mat}, t)$	Expected, fitness penalty at time period t during adulthood ($t > T_{ont}$)
$\pi(Y_{mat}, t)$	Expected fitness at time period t during adulthood ($t > T_{ont}$)
$\pi_{Total}(Y_{mat})$	Expected fitness across adulthood
π_0	Baseline fitness
$f(y)$	Mapping between phenotypic increments and fitness rewards (or penalties)
μ	Penalty weight

(ESM 3, lines 324-325)”

• You state that t runs from 0 to T and that t_{adult} runs from 0 to T_{adult} . But then in your notation $E0;t$ is not the same as $E0;t_{adult}$ when, say $t = t_{adult} = 3$. In other words $E0;3$ is not the same as $E0;3!!$ Of course many biologist may understand the sloppy notation, but that is not an excuse for using it. One way to avoid these problems is to have time run from time 0 (birth) to the end of the reproductive phase at time T_{end} . Maturity then occurs at time T , where $T + T_{adult} = T_{end}$.

REPLY: We agree with this helpful suggestion. We have adopted this way of indexing time throughout the ESM and refer to it in the main manuscript.

“

Environmental variable	Explanation
E_0	Environment 0
E_1	Environment 1
P_0	Optimal phenotype for E_0
P_1	Optimal phenotype for E_1
C_0	Cue indicating E_0
C_1	Cue indicating E_1
D_t	$D_t = \{x_1, x_2, \dots, x_t\}$, denotes the sequence of cues until time period t where x_1, x_2 , etc. until x_t denote the kind of cue (C_0 or C_1) sampled in each time period
t	Current time period ranges from $t = 0$ (birth) until T_{End} (the end of the reproductive cycle). It holds that $T_{End} = T_{Ont} + T_{Adult}$.
T_{ont}	Duration of ontogeny, i.e. ontogeny lasts for 10 time periods
T_{adult}	Duration of adulthood, i.e. adulthood lasts for 1, 5 or 10 time periods

(ESM 3, lines 225-226)”

“Thus, time runs from $t = 0$ (birth) until the end of the reproductive phase T_{end} , such that $T_{end} = T_{ont} + T_{adult}$. (lines 125-126)”